# The Impact of Grafting with Different Rootstocks on Eggplant (*Solanum melongena* L.) Growth and Its Rhizosphere Soil Microecology

Gongfu Du, Dan Zhu, Huang He, Xiaoliang Li, Yan Yang and Zhiqiang Qi *

Tropical Crops Genetic Resources Institute, Chinese Academy of Tropical Agricultural Sciences, Key Laboratory of Crop Gene Resources and Germplasm Enhancement in Southern China, Ministry of Agriculture and Rual Affairs, Key Laboratory of Tropical Crops Germplasm Resources Genetic Improvement and Innovation of Hainan Province, No. 4 Xueyuan Road, Longhua, Haikou 571101, China; dugongfu@catas.cn (G.D.); zhudan@catas.cn (D.Z.); huanghe@catas.cn (H.H.); xlli199777@163.com (X.L.); yziqi@126.com (Y.Y.)
* Correspondence: zhiqiangqi@catas.cn

**Abstract:** This study investigated the effects of grafting on eggplant growth, yield, and disease resistance, with a focus on microbial dynamics in the rhizosphere. Eggplant scions were grafted onto rootstocks of wild eggplant and tomato, with self-rooted eggplants serving as controls. Greenhouse experiments were conducted over an eight-month growing period, using standard field practices such as film mulching and integrated water–fertilizer management. High-throughput sequencing was used to analyze the biological properties and microbial community of the rhizosphere soil. Results showed that plants grafted onto 'Huimei Zhenba' and 'Torvum' rootstocks yielded up to 36.89% more than self-rooted controls, achieving yields of 4619.59 kg and 4399.73 kg per 667 m², respectively. The disease incidence of bacterial wilt was reduced to as low as 3.33% in the 'Huimei Zhenba' treatment, compared to 55.56% in non-grafted controls. Additionally, grafted plants exhibited increased stem diameter and chlorophyll content, with the TL/HM combination reaching 54.23 ± 3.17 SPAD units. The enhanced microbial biomass of carbon, nitrogen, and phosphorus, particularly in the TL/HM treatment (377.59 mg/kg, 28.31 mg/kg, and 36.30 mg/kg, respectively), supports a more nutrient-rich rhizosphere environment. Moreover, soil enzyme activities, such as β-glucosidase and phosphatase, were significantly higher in grafted plants, enhancing nutrient cycling and potentially increasing resistance to pathogens. Overall, grafted eggplants demonstrated enhanced soil microbial biomass, enzyme activity, and a more diverse microbial community, which are critical factors contributing to the improved yield and disease resistance observed in grafted crops.

**Keywords:** disease resistance; growth traits; soil microbe community; soil enzyme activity; yield



## 1. Introduction

Eggplant is a significant vegetable crop in China, covering approximately 866,700 hectares of cultivated land. It plays a crucial role in rural development and revitalization efforts [1]. However, to meet the increasing demand for eggplant, continuous cropping has become widespread. This has led to a rise in soil-borne diseases, significantly impacting crop health and yield [2]. One of the most severe threats is bacterial wilt, caused by *Ralstonia solanacearum*, which can cause widespread wilting and plant death, leading to substantial yield reductions. The typical incidence of bacterial wilt ranges from 20% to 30%, but in regions with prolonged eggplant cultivation or during severe outbreaks, infection rates can exceed 80%, causing devastating losses [3,4].

Over the years, efforts to control bacterial wilt in eggplants through biological agents, chemical treatments, and breeding for resistance have met with limited success. Recently, grafting onto superior rootstocks has become a widely recognized strategy in vegetable cultivation to enhance plant resilience against stress and disease, support growth and

development, and boost yield [5–7]. The use of rootstocks that are both highly compatible and resistant to disease is therefore crucial for optimizing eggplant production. Rootstock resistance to soil-borne diseases is not solely dependent on the genetic traits of the variety but is also influenced by the surrounding soil microecology [8,9]. Key factors and components of this environment include soil microbial communities, microbial biomass, and enzyme activity, all of which play critical roles in soil fertility, nutrient cycling, and overall plant growth and development [10–12].

Grafting tomatoes onto rootstocks is an effective strategy to enhance crop disease resistance. Significant variations were observed in seedling growth across different rootstocks, with coefficients of variation for leaf area and root dry matter at 18.24% and 15.21%, respectively. All rootstocks exhibited grafting survival rates above 90%. The average yield increase across treatments was 34.87%, with a yield coefficient of variation of 17.16%. Additionally, grafting significantly influenced the nutrient quality of tomato fruits, showing a high impact on vitamin C content (69.16% coefficient of variation) and a comparatively lower impact on soluble solids (8.87% coefficient of variation) [13]. Recent research indicates that resistant watermelon cultivars have significantly higher quantities of rhizosphere bacteria and actinomycetes, while displaying lower levels of rhizosphere fungi and *Fusarium*, compared to susceptible cultivars. The resistance of the watermelon cultivar is closely linked to the abundance of rhizosphere microorganisms. Grafting significantly increased the populations of rhizosphere bacteria and actinomycetes while reducing *Fusarium* levels in susceptible cultivars. Additionally, grafting altered the microbial community structure in the soil. These findings suggest that plant genotype has a substantial effect on soil microbial communities, and that variations in rhizosphere microbial composition may contribute to differences in resistance to *Fusarium oxysporum* f. sp. *niveum* among cultivars [14]. Grafting increased the levels of salicylic acid (SA), benzoic acid, vanillin, lignin, and polyamines (PAs), along with the activity of key enzymes, including phenylalanine ammonia lyase (PAL), polyphenol oxidase (PPO), and peroxidase (POD). These changes indicate that grafting enhanced pepper resistance to root rot. The disease incidence was reduced to 26.7% in grafted plants compared to 43.0% in non-grafted controls, while the disease index decreased to 11.7% in grafted plants versus 23.3% in controls [15].

Similarly, grafting peppers enhanced root health; the non-grafted (control) and grafted treatments showed disease incidences of 23.0% and 15.0%. After fifteen days post-inoculation, the root weight, length, surface area, volume, fork number, and tip number were 39.3%, 45.1%, 39.8%, 23.7%, 29.1%, and 36.5% higher for grafted plants than for the controls. These findings suggest that the grafted rootstock may be very useful for the prevention and control of bacterial wilt in cultivated peppers [16]. In addition, tomatoes grafted onto eggplant rootstocks showed a significant increase in cultivable bacteria, actinomycetes, microbial biomass (carbon, nitrogen, and phosphorus), and enzyme activities in the rhizosphere. The rhizosphere of grafted plants also exhibited greater bacterial richness, evenness, and diversity compared to self-rooted plants [17]. Previous studies have demonstrated that grafting eggplants onto different rootstocks can significantly improve yield and water use efficiency, particularly when using *Solanum torvum* under reduced irrigation conditions. Grafted plants exhibited increased plant height, leaf water potential, fruit size, and weight compared to controls, with *S. torvum* rootstocks showing superior performance. These grafted combinations also reduced water consumption while maintaining high productivity, suggesting grafting as an effective strategy for enhancing eggplant resilience and efficiency in water-limited environments [18]. Another study found that the *Beaufort/Black Bell* (Be/Bb) grafting combination promoted significant vegetative growth, with increased plant height, leaf number, and xylem vessel area compared to the non-grafted *Black Bell* (Bb) control, although some graft incompatibility issues were noted. Conversely, the *Solanum torvum/Black Bell* (To/Bb) grafts achieved a better balance between growth and yield, demonstrating improved fruit set and physiological stability. The combined effects of grafting and specific environmental conditions notably enhanced water uptake and overall plant performance [19]. These studies collectively illustrate the adaptability and effective-

ness of different rootstocks in improving grafted eggplant performance, particularly under challenging environmental conditions.

In addition to grafting, several factors may affect the yield and quality of eggplants, including genetics and cultivation practices. Eggplant yield and quality are shaped by genetic factors from both cultivated varieties and wild relatives. Key traits, such as fruit size, shape, and color, benefit from hybrid vigor (heterosis), which boosts yields in F1 hybrids. Wild relatives like *Solanum aethiopicum* and *S. linnaeanum* contribute resistance to Fusarium and Verticillium wilt, critical for yield stability [20,21]. Molecular techniques, including marker-assisted selection (MAS), streamline the incorporation of quantitative trait loci (QTL) linked to high phenolic and anthocyanin content, which enhance nutritional value and resilience [22,23]. Together, these genetic resources enable breeding programs to develop robust eggplant varieties capable of withstanding various environmental challenges while enhancing productivity and nutritional content.

The ecological factors affecting eggplant yield and quality were primarily managed through shade nets, improving eggplant yield and quality in Carnarvon's semi-arid environment [24]. The 21% shade net was optimal, reducing light intensity enough to prevent photoinhibition while sustaining photosynthesis, promoting taller, bushier plants with increased fruit yields [24]. Moderate shading (21% and 30%) significantly increased marketable fruit numbers, with minimal impact on total soluble solids and pH, proving that controlled shading effectively stabilizes yield and quality under intense light and temperature conditions [24]. In addition, the used of magnetized water (MW) irrigation significantly enhances soil conditions and eggplant yield and quality [25]. MW improved soil nutrient availability, increased microbial diversity, and boosted enzyme activities, which collectively fostered better plant growth. Eggplants under MW showed larger leaves, fruits, and higher chlorophyll content, contributing to increased yields. This sustainable practice enhances both soil health and crop performance, offering an efficient approach to boosting eggplant production through controlled irrigation [25]. Furthermore, a previous study emphasizes that weekly pruning, integrated pest management (IPM), and automated environmental controls (temperature, humidity, $CO_2$, and nutrients) significantly boost eggplant yield and quality in a high-tech glasshouse [26]. These practices ensure consistent water and nutrient delivery, reduce pest pressures, and provide optimal growth conditions, enhancing physiological performance and water use efficiency [26].

In addition to the grafting and parameters, the soil microorganisms play a critical role in resisting both biotic and abiotic stresses, with variations in their abundance and diversity having significant effects on crop growth, yield, soil health, and the occurrence and severity of soil-borne diseases [27]. Consequently, investigating shifts in soil microbial communities, microbial biomass, and enzyme activity within the rhizosphere of grafted eggplants is crucial for addressing the challenges posed by continuous cropping in eggplant cultivation [28].

The purpose of this study was to investigate the effects of grafting on plant growth, yield, and disease resistance, with a focus on the microbial dynamics in the rhizosphere soil. Specifically, the research aimed to compare the outcomes of grafting eggplant scions onto wild eggplant versus tomato rootstocks. Key factors considered included stem diameter, chlorophyll content, yield, and disease resistance. Additionally, the study examined changes in the diversity and composition of soil microbial communities, microbial biomass (carbon, nitrogen, and phosphorus), and soil enzyme activities (such as β-glucosidase, aminopeptidases, and phosphatases). The research also explored correlations between these microecological factors and plant performance to elucidate the mechanisms by which grafting enhances resistance to soil-borne pathogens and improves crop yield.

## 2. Materials and Methods

### 2.1. Experimental Materials

The experimental materials comprised eggplant-type rootstocks 'Huimei Zhenba', *S. torvum* (referred to as 'Torvum'), and 'Beike', along with tomato-type rootstocks 'Qian-

gli' and 'Saint Nise'. The eggplant cultivar utilized in the study was 'Tianlong No. 9', a purple long eggplant cultivar. The rootstocks 'Huimei Zhenba', 'Torvum', and 'Beike' were obtained from the perennial vegetable germplasm collection of Hainan Province (109.29′27.92″ E, 19.29′24.35″ N), while the other seeds were sourced from the vegetable seed market in Haikou City, Hainan Province (110.19′23.74″ E, 20.2′36.42″ N). The experiment was conducted from October 2021 to May 2022 at the field observation station of the Danzhou Vegetable Field, Chinese Academy of Tropical Agricultural Sciences (109.29′47.40″ E, 19.30′25.92″ N).

### 2.2. Experimental Design

Six treatments were established, with 'Tianlong No. 9' eggplant grafted onto five different rootstocks: ① 'Tianlong No. 9/Huimei Zhenba' (TL/HM), ② 'Tianlong No. 9/Torvum' (TL/TM), ③ 'Tianlong No. 9/Beike' (TL/BK), ④ 'Tianlong No. 9/Qiangli' (TL/QL), and ⑤ 'Tianlong No. 9/Saint Nise' (TL/SNS). The sixth treatment consisted of self-rooted 'Tianlong No. 9/Tianlong No. 9' eggplant plants (TL/TL) serving as the control. Each treatment included 30 plants, arranged in a double-row bed with a plant spacing of 50 cm and a row spacing of 50 cm, covering an area of 9 square meters per plot. Each treatment consisted of 30 plants of a given cultivar arranged in a 9-square-meter plot, which was considered one replicate. A total of three replicates were used per treatment, resulting in 90 plants per treatment (3 replicates × 30 plants per replicate). The experiment utilized greenhouse plug tray seedling cultivation. The test rootstocks (Huimei and Torvum) were sown on 8 October 2021, while the other rootstocks (Beike, Qiangli, and Saint Nise) and scion (Tianlong No. 9 eggplant) were sown on 30 October. Self-rooted seedlings were sown on November 21. Grafting was conducted on 6 December, when the rootstock diameter was between 2.5 and 3.0 mm and the scion had 4–5 leaves. Healthy seedlings were transplanted 21 days after grafting. The grafting and subsequent transplanting followed standard procedures, including the use of film covering and standard field management. Figure 1 presents photos of the different graft combinations at the experimental site: TL/TM (Tianlong No. 9/Torvum), TL/HM (Tianlong No. 9/Huimei Zhenba), TL/BK (Tianlong No. 9/Beike), TL/TL (Tianlong No. 9/Tianlong No. 9), TL/QL (Tianlong No. 9/Qiangli), and TL/SNS (Tianlong No. 9/Saint Nise).

### 2.3. Experimental Conditions

The experiment was conducted from October 2021 to May 2022 at the Danzhou Caitian Field Observation and Experimental Station of the Chinese Academy of Tropical Agricultural Sciences (109.29′47.40″ E, 19.30′25.92″ N). The area experiences an average annual temperature of 22–27 °C, with 1750–2650 h of sunshine per year, yielding a sunshine rate of 50–60%. The climate provides ample light and warmth throughout the year.

The region receives an average annual rainfall of 1639 mm, with distinct wet and dry seasons. The rainy season, from May to October, accounts for approximately 1500 mm (70–90%) of the annual precipitation, while the drier season, from November to April, contributes only 10–30% of the total rainfall. Table 1 summarizes the average annual temperature (°C) (Figure 2A) and annual rainfall (mm) (Figure 2B) at the experimental site, recorded from 2016 to 2022 at the Danzhou Caitian Field Observation and Experimental Station.

The soil at the experimental site is classified as typical brick-red soil, managed under a continuous eggplant-summer fallow rotation system. Prior to the experiment, the basic physical and chemical properties of the 0–20 cm soil layer were as follows: organic matter, 16.54 g·kg$^{-1}$; alkaline nitrogen (AN), 57.51 mg·kg$^{-1}$; available phosphorus (AP), 57.14 mg·kg$^{-1}$; available potassium (AK), 157.08 mg·kg$^{-1}$; and pH, 5.4.

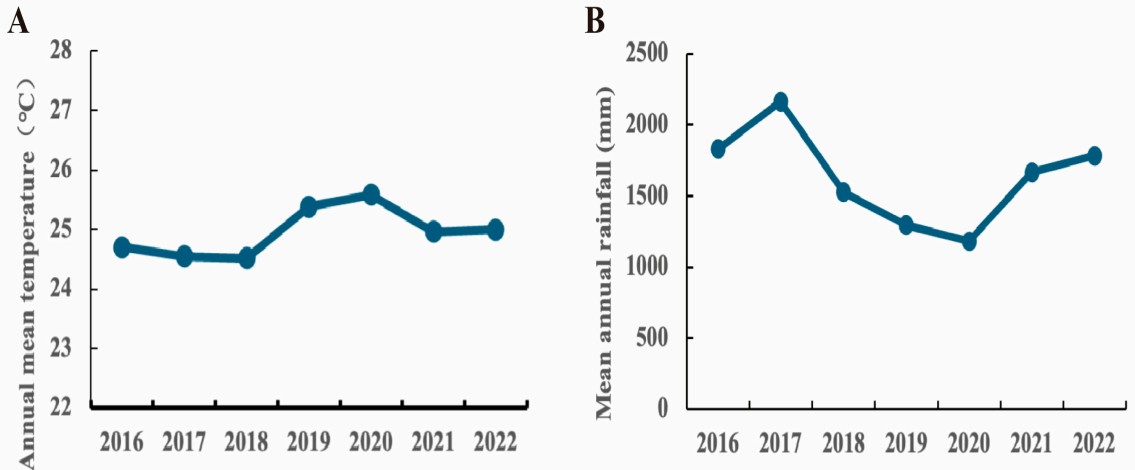

**Figure 1.** The different graft combinations at the experimental site. (**A**) TL/TM (Tianlong No. 9/Torvum). (**B**) TL/HM (Tianlong No. 9/Huimei Zhenba). (**C**) TL/BK (Tianlong No. 9/Beike). (**D**) TL/TL (Tianlong No. 9/Tianlong No. 9). (**E**) TL/QL (Tianlong No. 9/Qiangli). (**F**) TL/SNS (Tianlong No. 9/Saint Nise).

**Figure 2.** The average annual temperature (°C) and annual rainfall (mm) at the experimental site were recorded from 2016 to 2022 at the Danzhou Caitian Field Observation and Experimental Station. (**A**) Average mean temperature (°C). (**B**) Mean annual rainfall (mm).

**Table 1.** The average annual temperature (°C) and annual rainfall (mm) at the experimental site were recorded from 2016 to 2022 at the Danzhou Caitian Field Observation and Experimental Station.

| Year | 2016 | 2017 | 2018 | 2019 | 2020 | 2021 | 2022 |
|---|---|---|---|---|---|---|---|
| Average annual temperature (°C) | 24.70 | 24.55 | 24.52 | 25.38 | 25.58 | 24.96 | 24.00 |
| Annual rainfall (mm) | 1831.0 | 2164.0 | 1525.0 | 1293.5 | 1180.1 | 1666.9 | 1783.2 |

To control pests such as thrips and whiteflies, yellow and blue insect traps were deployed at a density of 300 sheets/hm². For fertilization, the base fertilizer included 6000 kg/hm² of commercial sheep manure organic fertilizer (containing $\geq$ 30% organic matter and N + $P_2O_5$ + $K_2O$ $\geq$ 4%) and 375 kg/hm² of compound fertilizer (N:$P_2O_5$:$K_2O$ = 15:15:15). The primary topdressing schedule was as follows: (1) seven days post-transplantation, 45 kg/hm² of compound fertilizer (N:$P_2O_5$:$K_2O$ = 15:15:15) was applied; (2) during flowering, 75 kg/hm² of compound fertilizer (N:$P_2O_5$:$K_2O$ = 15:15:15) was added; (3) during the initial fruiting and fruit expansion periods, 75 kg/hm² of water-soluble and compound fertilizer (N:$P_2O_5$:$K_2O$ = 15:5:25) was applied; and (4) during the peak fruiting period, 120 kg/hm² of water-soluble and compound fertilizer (N:$P_2O_5$:$K_2O$ = 15:5:25) was applied. Additionally, foliar fertilizers, including 0.3% potassium dihydrogen phosphate and boron fertilizer, were applied from the flowering to the fruiting stages to enhance fruit set rates.

*2.4. Statistical Analysis*

Data collected during the experiment were analyzed using Duncan's new multiple range test to evaluate statistical significance, and the results were subsequently visualized for better interpretation.

*2.5. Determination of Growth Traits of Eggplant Grafting with Different Rootstocks*

During the peak fruit-bearing period of eggplants, key traits of both grafted and self-rooted seedlings were assessed, with 12 plants evaluated per treatment group. These 12 plants were selected from each plot (which consisted of 30 plants) using an S-type sampling method. Plant height was measured using a steel tape, from the uppermost heart leaf to ground level, representing the mature plant height. The scion diameter (measured 1.5 cm above the grafting point) and rootstock stem diameter (measured 1.5 cm below the grafting point) were determined using a vernier caliper. Additionally, the chlorophyll content was quantified using a TYS-4N plant nutrition analyzer (Top Cloud-Agri, Hangzhou, China).

*2.6. Determination of Different Rootstocks on Disease Resistance and Yield of Eggplant*

The field disease investigation was conducted 90 days after transplanting. For each treatment, the disease incidence of all plants was recorded, and the disease incidence rate of the grafted eggplant seedlings was calculated using the following formula:

$$\text{Disease incidence rate (\%)} = (\text{Number of diseased plants} / \text{Total number of plants investigated}) \times 100\%.$$

The criteria for evaluating diseased plants included observable symptoms such as leaf and branch wilting, stem desiccation and discoloration, and the inability of the plant stem to remain upright [29].

The yield data in this study are presented as the average yield per individual plant. During the fruit harvest period, the total yield of eggplant plants within the experimental plots was recorded, and the average yield per plant was calculated. The final harvest occurred on 12 May 2022.

### 2.7. Determination of Soil Microbial Biomass (Carbon, Nitrogen, and Phosphorus)

Field soil samples were collected using the S-type sampling method, with three soil samples taken for each treatment. The soil samples ware passed through a 2 mm sieve and stored at 4 °C for the determination of soil microbial biomass. Five grams of fresh soil were weighed for microbial biomass carbon (C) and nitrogen (N) analysis, and 3 g were weighed for microbial biomass phosphorus (P). Each sample ($n = 3$ for each treatment) was placed into a 100 mL beaker. The beakers were then placed in a desiccator containing a few sheets of water-moistened filter paper at the bottom. Additionally, a small beaker containing 50 mL of NaOH solution and another small beaker with approximately 50 mL of ethanol-free chloroform (with a small amount of anti-bumping granules) were placed inside the desiccator. The desiccator was sealed with a small amount of petroleum jelly, and a vacuum pump was used to evacuate the air until the chloroform began to boil, maintaining this for at least 2 min. The desiccator valve was then closed, and the samples were stored in the dark at 25 °C for 24 h. The beakers containing the NaOH solution and chloroform were removed, and the chloroform was returned to its bottle for reuse.

For the extraction of microbial biomass carbon and nitrogen, the entire soil sample was transferred into a 250 mL Erlenmeyer flask, and 25 mL of potassium sulfate solution was added. The mixture was then extracted by shaking on an oscillator for 30 min before being filtered. At the beginning of the fumigation process, an equal amount of soil was weighed and extracted using potassium sulfate solution as described above. The extract was either measured immediately or stored at −15 °C. The carbon (C) and nitrogen (N) contents of the extract were then directly determined using a carbon–nitrogen analyzer.

For the extraction of microbial biomass phosphorus, after fumigation, the entire soil sample was transferred into a 250 mL Erlenmeyer flask, and 50 mL of sodium bicarbonate solution was added. The mixture was then extracted by shaking on an oscillator for 30 min before being filtered. At the start of fumigation, an equal amount of soil was weighed and extracted using sodium bicarbonate solution as described above. The extract was either measured immediately or stored at −15 °C. Separately, 3 g of fresh soil was taken, and 0.5 mL of a 250 mg/L phosphorus standard solution along with 50 mL of sodium bicarbonate solution was added. This mixture was also extracted by shaking on an oscillator for 30 min before being filtered. This extract was used to determine the recovery rate (Rp) of the added orthophosphate inorganic salt, which served to correct for the soil's adsorption and fixation of microbial biomass phosphorus released during fumigation.

### 2.8. Determination of Soil Enzyme Activity (β-Glucosidase, Aminopeptidase, and Phosphatase)

Field soil samples were collected using the S-type sampling method, with three soil samples taken for each treatment. The soil samples ware passed through a 2 mm sieve and stored at 4 °C for the determination of soil enzyme activity. For the assessment of β-glucosidase activity, the fresh soil samples were either allowed to air-dry naturally or dried in an oven at 37 °C, after which they were sieved through a 30–50 mesh screen. A 0.1 g portion of the soil sample ($n = 3$ for each treatment) was then taken and mixed with 50 μL of toluene, ensuring thorough mixing so that the soil was fully moistened. The mixture was left to stand at room temperature for 15 min. Subsequently, 500 μL of Reagent 1 and 100 μL of Reagent 2 from the kit were added to the soil sample, and the mixture was incubated at 37 °C for 3 h. Immediately following incubation, the mixture was boiled in a water bath for 5 min, with the container tightly covered to prevent water loss, and then cooled under running water. The mixture was then centrifuged at 10,000 rpm for 10 min at room temperature, and the supernatant was collected for further analysis. A 100 μL aliquot of the supernatant was taken and mixed with 900 μL of Reagent 4. The mixture was allowed to stand for 10 min before measuring the absorbance at 400 nm using a full-wavelength microplate reader (Molecular Devices, San Jose, CA, USA), with distilled water as the blank and a 1 cm light path.

For the assessment of aminopeptidase activity, the soil samples were allowed to air-dry naturally and then sieved through a 30–50 mesh screen. A 0.1 g portion of the air-dried soil

sample ($n$ = 3 for each treatment) was taken, to which 15 μL of toluene was added. The mixture was shaken thoroughly and left to stand at room temperature for 15 min. Following this, 255 μL of Reagent 1 and 30 μL of Reagent 2 were added. The mixture was incubated in a water bath at 30 °C for 1 h and then immediately boiled for 5 min. After boiling, the mixture was cooled to room temperature under running water. It was then centrifuged at $14,000 \times g$ for 10 min at room temperature. A 200 μL aliquot of the supernatant was taken, and its absorbance was measured at 405 nm using a full-wavelength microplate reader (Molecular Devices, San Jose, CA, USA).

For the assessment of phosphatase activity, the fresh soil samples were allowed to air-dry naturally or were dried in an oven at 37 °C, and then sieved through a 30–50 mesh screen. A 0.1 g portion of the soil sample ($n$ = 3 for each treatment) was taken, and 50 μL of toluene was added. The mixture was thoroughly shaken to ensure the soil was fully moistened and then left to stand at room temperature for 15 min. Subsequently, 250 μL of Reagent 1 and 250 μL of Reagent 2 were added, and the mixture was thoroughly mixed before being incubated at 37 °C for 24 h. After incubation, 500 μL of Reagent 3 was added and mixed well. The mixture was then centrifuged at 10,000 rpm for 10 min at room temperature, and the supernatant was collected for further analysis. A 100 μL aliquot of the supernatant was taken, and 700 μL of Reagent 4, 100 μL of Reagent 5, and 100 μL of Reagent 6 were sequentially added, with thorough mixing after each addition. The mixture was allowed to stand for 10 min, and the absorbance was measured at 570 nm using a full-wavelength microplate reader with a 1 cm light path.

*2.9. Soil Microbial Sequencing and Analysis*

The soil samples ware passed through a 2 mm sieve and stored at −80 °C for subsequent analysis of soil microbial community structure and diversity. DNA from each sample were extracted using the FastPure Soil DNA isolation Kit (MJYH, Shanghai, China) in accordance with the manufacturer's instructions. After completing the extraction of genomic DNA, the quality of the extracted DNA was assessed using 1% agarose gel electrophoresis. For library construction, adapter sequences were added to the ends of the target regions via PCR. The PCR products were purified using the AxyPrep DNA Gel Extraction Kit (Axygen, Corning, NY, USA), eluted with Tris-HCl buffer, and analyzed using 2% agarose gel electrophoresis. Denaturation was performed using sodium hydroxide, and single-stranded DNA fragments were generated with the TruSeq™ DNA Sample Prep Kit (Illumina, San Diego, CA, USA). For the sequencing, the samples were sequenced using the NextSeq 2000 platform (Illumina, San Diego, CA, USA) which was conducted by MajorBio (MajorBio, Shanghai, China).

Optimized sequences were extracted to remove redundancy, thereby reducing computational overhead during analysis (http://drive5.com/usearch/manual/dereplication.html) (accessed date: 25 April 2023). Single sequences without duplicates were excluded (http://drive5.com/usearch/manual/singletons.html) (accessed date: 28 April 2023). The non-redundant sequences (excluding single sequences) were clustered into OTUs at a 97% similarity threshold, with chimeras removed during clustering to obtain representative OTU sequences. All optimized sequences were mapped to these OTU representative sequences, and sequences with ≥97% similarity to the OTU representative sequences were used to generate the OTU table.

To determine the taxonomic information corresponding to each OTU, the RDP classifier Bayesian algorithm was employed for taxonomic analysis [30] of the OTU representative sequences at the 97% similarity level. The community composition of each sample was then statistically analyzed across various taxonomic levels: domain, kingdom, phylum, class, order, family, genus, and species. The following databases were used for comparison: the 16S rRNA Bacteria and Archaea Ribosomal Database (defaulting to the Silva database unless otherwise specified): Silva (Release 138, http://www.arb-silva.de) (accessed date: 3 May 2023) [31]; RDP (Release 11.5, http://rdp.cme.msu.edu/) (accessed date: 3 May 2023); and Greengenes (Release 135, http://greengenes.secondgenome.com/) (accessed date:

3 May 2023) [32]. For the ITS Fungal Database, Unite was used (Release 8.0, https://unite.ut.ee/) (accessed date: 10 May 2023) [33]. Functional gene analysis was conducted using the FGR, an RDP-curated functional gene database derived from GenBank (Release 7.3, http://www.fungene-db.fr/) (accessed date: 15 May 2023). The software and algorithms utilized included the Qiime platform (version 1.9.1) (http://qiime.org/scripts/assign_taxonomy.html) (accessed date: 17 May 2023) [34] and the RDP Classifier (version 2.11, http://sourceforge.net/projects/rdp-classifier/) (accessed date: 20 May 2023), with a confidence threshold set at 0.7.

Through the analysis of Alpha diversity indices, information regarding species richness, diversity, and coverage within a community was obtained. Group difference tests were conducted to determine whether the Alpha diversity index values between two or more groups exhibited significant differences. The methods used for testing group differences included one-way ANOVA for multiple group comparisons and the Kruskal–Wallis rank-sum test; for two-group comparisons, the student's *t*-test, Welch's *t*-test, and the Wilcoxon rank-sum test were employed.

For the soil microbial comparison between two groups, the Student's *t*-test, also known as the *t*-test, utilized the theory of t-distribution to infer the probability of observed differences, thereby determining whether the difference between two means was significant. This test was primarily applied to normally distributed data with a small sample size ($n < 30$) where the population standard deviation ($\sigma$) was unknown. Welch's *t*-test was selected when the variances between two groups were unequal. The Wilcoxon rank-sum test, also known as the Mann–Whitney U test, was used as a non-parametric method for comparing two independent samples.

## 3. Results

### 3.1. Effects of Grafting with Different Rootstocks on the Growth Traits of Eggplant

To elucidate the effects of grafting with different rootstocks on the growth traits of eggplant, we conducted grafting using five distinct rootstocks and monitored several parameters, including plant height, rootstock stem diameter, scion stem diameter, and chlorophyll content. The results indicated that grafting effectively enhanced both scion stem diameter and chlorophyll content. In terms of plant height, the treatments utilizing eggplant rootstocks (TL/HM, TL/BK, and TL/TM) produced the highest plants, measuring 98.50 cm, 98.75 cm, and 99.17 cm, respectively (Figure 3A). Although there were no significant differences among these three treatments, they were significantly higher than the self-rooted eggplant plants (TL/TL), which had a height of 94.92 cm. No significant difference was observed between the SNS treatment, which used tomato rootstock, and the TL/TL control, while the TL/QL treatment exhibited a significant difference.

Regarding rootstock stem diameter, the TL/HM treatment resulted in the best diameter, with a diameter of $2.49 \pm 0.24$ cm. This was not significantly different from the TL/TM treatment but was significantly better than the other rootstock treatments. There were no significant differences in stem diameter among the TL/BK, TL/QL, and TL/SNS treatments (Figure 3B).

For scion stem diameter, all grafted plants displayed thicker stems compared to the self-rooted eggplant plants, with significant differences observed between the grafted treatments and the TL/TL control. However, no significant differences were found among the various grafted treatments (Figure 3C).

Concerning chlorophyll content, all grafted eggplant treatments demonstrated higher chlorophyll levels compared to the TL/TL control. The TL/HM treatment exhibited the highest chlorophyll content, reaching $54.23 \pm 3.17$ SPAD, which was significantly different from the TL/TL control but not significantly different from the other grafted treatments (Figure 3D).

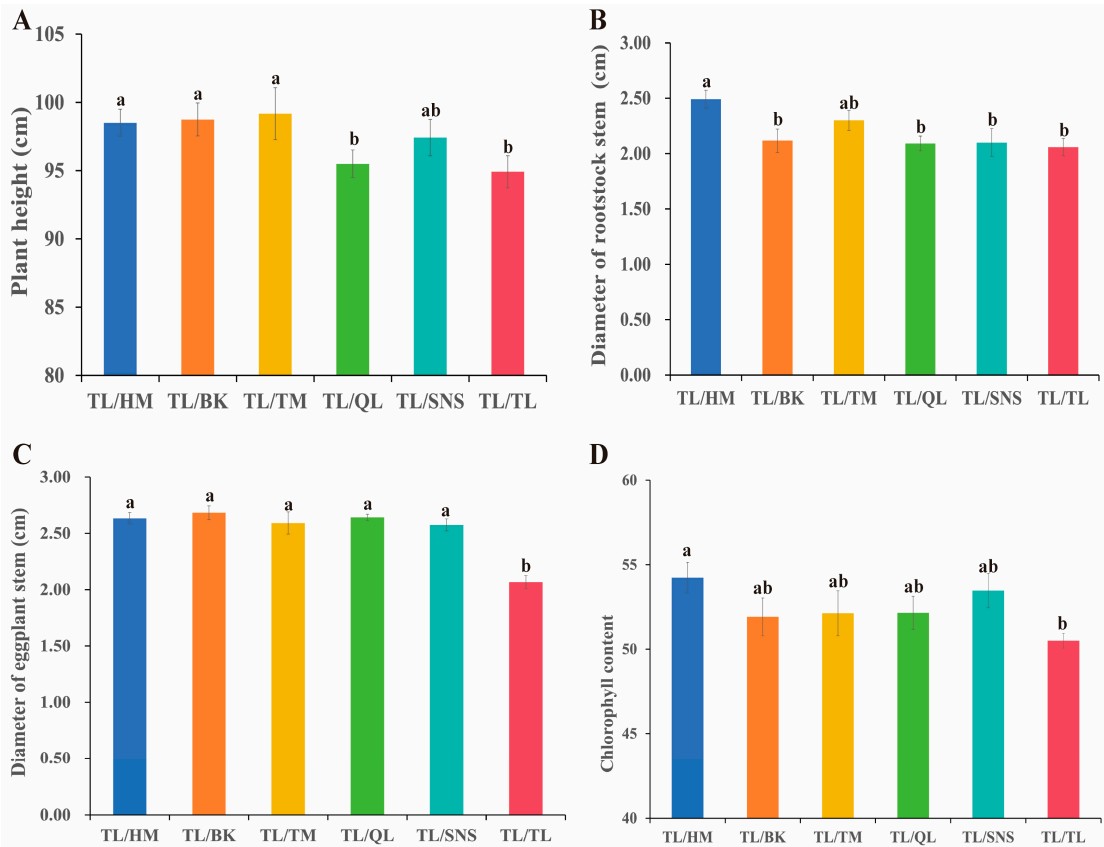

**Figure 3.** The effects of grafting with different rootstocks on the growth traits of eggplant. (**A**) The plant height after grafting with different rootstocks. (**B**) The diameter of rootstock stem after grafting with different rootstocks. (**C**) The diameter of eggplant stem after grafting with different rootstocks. (**D**) The chlorophyll content after grafting with different rootstocks. Error bars represent the standard error of the mean (SEM), and statistical differences among tissues ($p < 0.05$) are denoted by different lowercase letters. Indicator: TL/HM, Tianlong No. 9/Huimei Zhenba; TL/BK, Tianlong No. 9/Beike; TL/TM, Tianlong No. 9/Torvum; TL/QL, Tianlong No. 9/Qiangli; TL/SNS, Tianlong No. 9/Saint Nise; TL/TL, Tianlong No. 9/Tianlong No. 9.

*3.2. Effects of Different Rootstocks on Disease Resistance and Yield of Eggplant*

The field disease investigation was conducted 90 days after transplanting. For each treatment, the disease incidence of all plants was recorded, and the disease incidence rate of the grafted eggplant seedlings was calculated. Based on the results, TL/HM has the lowest disease incidence rate at 8.33%, indicating the highest resistance to bacterial wilt. TL/BK shows the highest disease incidence at 30.00%, suggesting this rootstock is the least resistant among the five cultivars. TL/TM has a moderately high disease incidence of 20.00%. TL/QL and TL/SNS have disease incidence rates of 15.00% and 11.67%, respectively, indicating they perform better than TL/BK and TL/TM but not as well as TL/HM.

As illustrated in Figure 4, grafting eggplants onto rootstocks significantly reduced the incidence of bacterial wilt. The lowest disease incidence was observed in the TL/HM, TL/SNS, and TL/TM treatments, with rates of 3.33%, 5.56%, and 7.78%, respectively (Figure 4A). These were followed by the TL/BK and TL/QL treatments, which had incidence rates of 25.56% and 32.22%, respectively. The highest disease incidence was recorded in the self-rooted plants (TL/TL), at 55.56%.

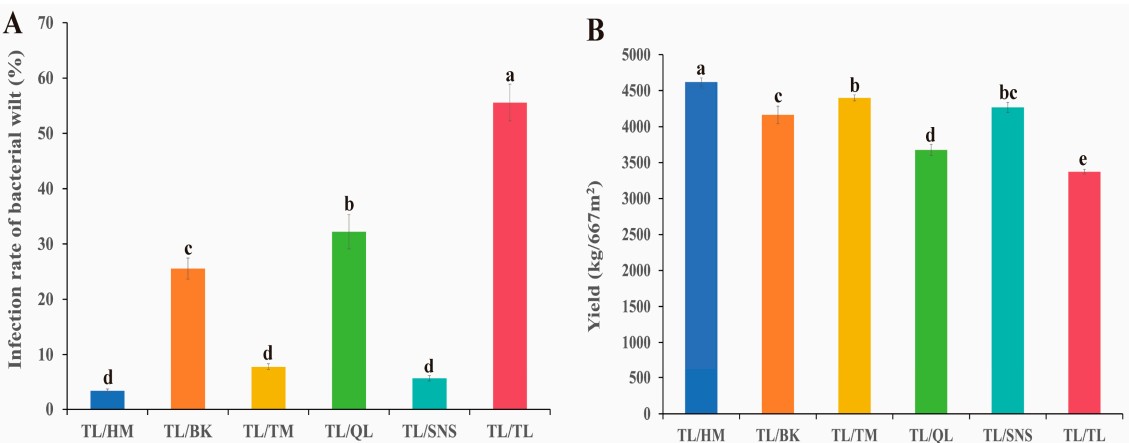

**Figure 4.** The effects of different rootstocks on disease resistance and yield of eggplant. (**A**) The infection rate of bacterial wilt after grafting with different rootstocks. (**B**) The yield of eggplants after grafting with different rootstocks. Error bars represent the standard error of the mean (SEM), and statistical differences among tissues ($p < 0.05$) are denoted by different lowercase letters. Indicator: TL/HM, Tianlong No. 9/Huimei Zhenba; TL/BK, Tianlong No. 9/Beike; TL/TM, Tianlong No. 9/Torvum; TL/QL, Tianlong No. 9/Qiangli; TL/SNS, Tianlong No. 9/Saint Nise; TL/TL, Tianlong No. 9/Tianlong No. 9.

Grafting with rootstocks has been shown to significantly enhance eggplant yield, with all grafted treatments achieving higher yields than self-rooted eggplants (TL/TL). Among the grafted treatments, the TL/HM treatment exhibited the highest yield per plant, reaching 2107.22 g, followed by the SNS and TL/TM treatments, with yields of 1963.33 g and 1923.33 g per plant, respectively. The yields per plant for the TL/BK, TL/QL, and self-rooted (TL/TL) plants were 1872.22 g, 1740.00 g, and 1522.22 g, respectively. Grafting resulted in yield increases ranging from 14.31% to 38.43% compared to self-rooted plants.

The increase in yield due to grafting was also evident, with all grafted treatments producing significantly higher yields compared to the self-rooted eggplants (TL/TL). The TL/HM treatment achieved the highest yield, reaching 4619.59 kg per 667 m$^2$, which was significantly better than the yields obtained from the other treatments (Figure 4B). The TL/TM treatment followed, with a yield of 4399.73 kg per 667 m$^2$. The yields for TL/BK, TL/SNS, TL/QL, and TL/TL were 4162.57 kg per 667 m$^2$, 4266.33 kg per 667 m$^2$, 3675.91 kg per 667 m$^2$, and 3374.53 kg per 667 m$^2$, respectively. The yield increase resulting from grafting, compared to self-rooted plants, ranged from 8.93% to 36.89%.

*3.3. Soil Microbial Biomass Analysis*

The microbial biomass content of carbon, nitrogen, and phosphorus in the rhizosphere soil of grafted plants was significantly higher than that observed in self-rooted eggplants (Table 2). Among the various treatments, TL/HM and TL/SNS demonstrated the most favorable results, with notable increases in microbial biomass carbon (TL/HM: 377.59 mg/kg; TL/SNS: 379.91 mg/kg), nitrogen (TL/HM: 28.31 mg/kg; TL/SNS: 27.17 mg/kg), and phosphorus (TL/HM: 36.30 mg/kg; TL/SNS: 36.74 mg/kg) when compared to the other grafted treatments and highly significant differences compared to the self-rooted eggplants. The TL/TM treatment also showed significant increases in microbial biomass phosphorus, particularly when compared to the TL/BK and TL/QL treatments, and highly significant differences relative to the self-rooted eggplants.

**Table 2.** The soil microbial biomass analysis of carbon, nitrogen, and phosphorus.

|  | **Carbon** | **Nitrogen** | **Phosphorus** |
|---|---|---|---|
| TL/HM | 377.59 ± 12.45 [a] | 28.31 ± 0.92 [a] | 36.30 ± 0.72 [a] |
| TL/BK | 343.28 ± 15.24 [b] | 22.67 ± 1.09 [b] | 27.64 ± 0.83 [c] |
| TL/TM | 348.04 ± 5.69 [b] | 23.14 ± 0.24 [b] | 30.30 ± 0.97 [b] |
| TL/QL | 347.59 ± 10.19 [b] | 23.06 ± 1.37 [b] | 28.44 ± 1.01 [c] |
| TL/SNS | 379.64 ± 13.14 [a] | 27.17 ± 0.85 [a] | 36.74 ± 1.09 [a] |
| TL/TL | 294.91 ± 10.89 [c] | 20.48 ± 1.07 [c] | 20.44 ± 1.11 [d] |

Statistical differences among tissues ($p < 0.05$) are denoted by different lowercase letters. Indicator: TL/HM, Tianlong No. 9/Huimei Zhenba; TL/BK, Tianlong No. 9/Beike; TL/TM, Tianlong No. 9/Torvum; TL/QL, Tianlong No. 9/Qiangli; TL/SNS, Tianlong No. 9/Saint Nise; TL/TL, Tianlong No. 9/Tianlong No. 9.

*3.4. Soil Enzymes Activity Assessment*

The activities of β-glucosidase, aminopeptidase, and phosphatase in the rhizosphere soil of grafted plants were significantly higher than those observed in the rhizosphere soil of self-rooted eggplants. Specifically, β-glucosidase activity in the rhizosphere soil of grafted plants was markedly elevated compared to self-rooted eggplants, with the highest levels recorded in the TL/HM and TL/SNS treatments, reaching 79.84 µg/g.h and 79.19 µg/g.h, respectively (Table 3). There was no significant difference between these two treatments, but both showed significant differences compared to the other treatments. Aminopeptidase activity was similarly elevated in the TL/HM and TL/SNS grafted plants, measuring 4.15 U/g dry sample and 3.97 U/g dry sample, respectively (Table 3). Again, there was no significant difference between these two treatments, but they were significantly different from the other treatments. Phosphatase activity also peaked in the TL/HM and TL/SNS grafted plants, at 3.45 mg/g.24 h and 3.38 mg/g.24 h, respectively, with no significant difference between the two, but a significant difference compared to the other treatments (Table 3). These results suggest that grafting enhances the activity of enzymes involved in the soil carbon, nitrogen, and phosphorus cycles within the rhizosphere soil of grafted plants, thereby promoting material cycling within the rhizosphere microenvironment.

**Table 3.** The soil enzyme activity analysis of β-glucosidase, aminopeptidase, and phosphatase.

|  | **β-Glucosidase** | **Aminopeptidase** | **Phosphatase** |
|---|---|---|---|
| TL/HM | 79.84 ± 2.46 [a] | 4.15 ± 0.08 [a] | 3.45 ± 0.13 [a] |
| TL/BK | 56.53 ± 1.74 [c] | 3.43 ± 0.12 [bc] | 3.16 ± 0.10 [c] |
| TL/TM | 75.45 ± 1.94 [b] | 3.56 ± 0.13 [b] | 3.22 ± 0.08 [bc] |
| TL/QL | 54.88 ± 1.68 [c] | 3.27 ± 0.08 [c] | 3.16 ± 0.11 [c] |
| TL/SNS | 79.19 ± 2.08 [a] | 3.97 ± 0.13 [a] | 3.38 ± 0.03 [ab] |
| TL/TL | 49.26 ± 2.53 [d] | 3.01 ± 0.10 [d] | 2.94 ± 0.05 [d] |

Statistical differences among tissues ($p < 0.05$) are denoted by different lowercase letters. Indicator: TL/HM, Tianlong No. 9/Huimei Zhenba; TL/BK, Tianlong No. 9/Beike; TL/TM, Tianlong No. 9/Torvum; TL/QL, Tianlong No. 9/Qiangli; TL/SNS, Tianlong No. 9/Saint Nise; TL/TL, Tianlong No. 9/Tianlong No. 9.

*3.5. Analysis of the Alpha Diversity of Rhizosphere Soil Microbial Communities in Grafted Plants*

As illustrated in Figure 5, the Ace, Chao, and Shannon indices of bacterial communities in the rhizosphere soil of the TL/HM (Ace: 7.46%; Chao: 7.51%; and Shannon: 19.46%) and TL/SNS (Ace: 9.19%; Chao: 9.03%; and Shannon: 20.62%) grafted plants were significantly higher than those observed in the other treatments, whereas the Simpson index of TL/HM (−57.97%) and TL/SNS (−73.91%) was significantly lower compared to the TL/BK (−23.19%), TL/TM (−39.13%), TL/QL (−23.19%), and TL/TL treatments. The sequencing library coverage for each sample exceeded 98%, confirming that the results accurately reflect the microbial communities present in the samples. The analysis of the Ace (Figure 5A), Chao 1 (Figure 5B), Shannon (Figure 5C), and Simpson (Figure 5D) indices revealed that, compared to self-rooted eggplants, the grafting treatments significantly increased bacterial diversity in the rhizosphere soil, with the TL/SNS and TL/HM treatments

showing the most pronounced effects. Additionally, grafting was shown to significantly enhance bacterial richness.

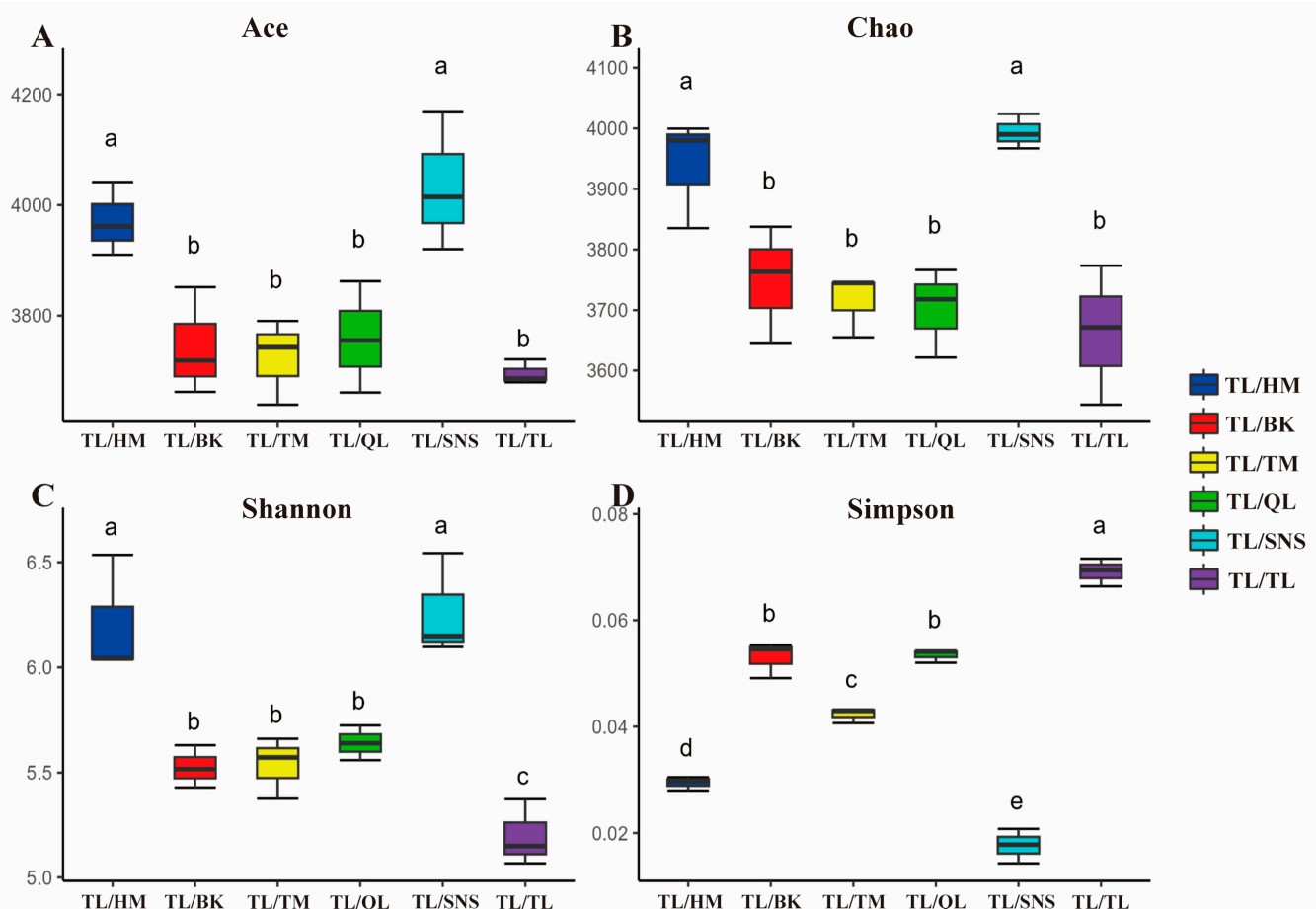

**Figure 5.** The soil bacterial diversity index in rhizosphere between grafting and non-grafting plants. (**A**) The Ace indices of soil bacterial diversity. (**B**) The Chao indices of soil bacterial diversity. (**C**) The Shannon indices of soil bacterial diversity. (**D**) The Simpson index of soil bacterial diversity. The statistical differences among tissues ($p < 0.05$) are denoted by different lowercase letters. Indicator: TL/HM, Tianlong No. 9/Huimei Zhenba; TL/BK, Tianlong No. 9/Beike; TL/TM, Tianlong No. 9/Torvum; TL/QL, Tianlong No. 9/Qiangli; TL/SNS, Tianlong No. 9/Saint Nise; TL/TL, Tianlong No. 9/Tianlong No. 9.

The Ace (Figure 6A) and Chao (Figure 6B) indices of fungal communities in the rhizosphere soil of the TL/SNS (Ace: 25.44% and Chao: 31.02%) and TL/HM (Ace: 17.77% and Chao: 25.26%) grafted plants were significantly higher than those observed in the TL/BK (Ace: 7.52% and Chao: 10.48%), TL/TM (Ace: 0.62% and Chao: 4.53%), TL/QL (Ace: −2.31% and Chao: 3.74%), and TL/TL plants. In terms of the Shannon index, the TL/SNS (38.38%), TL/HM (37.82%), and TL/TM (23.23%) grafted treatments exhibited significantly higher values compared to the TL/BK (4.62%), TL/QL (4.89%), and TL/TL plants, though there were no significant differences among the three grafted treatments (Figure 6C). The Simpson index was significantly lower in the TL/SNS (−41.81%) and TL/HM (−41.66%) treatments compared to the TL/BK (−15.11%), TL/QL (−12.81%), and TL/TL plants (Figure 6D). The sequencing library coverage for each sample exceeded 98%, ensuring that the results accurately reflect the microbial communities in the samples. The analysis of the Ace, Chao 1, Shannon, and Simpson indices revealed that, compared to self-rooted eggplants, grafting treatments significantly increased the diversity of fungal communities

in the rhizosphere soil, with TL/SNS and TL/HM showing the most pronounced effects. Additionally, grafting was found to significantly enhance fungal richness.

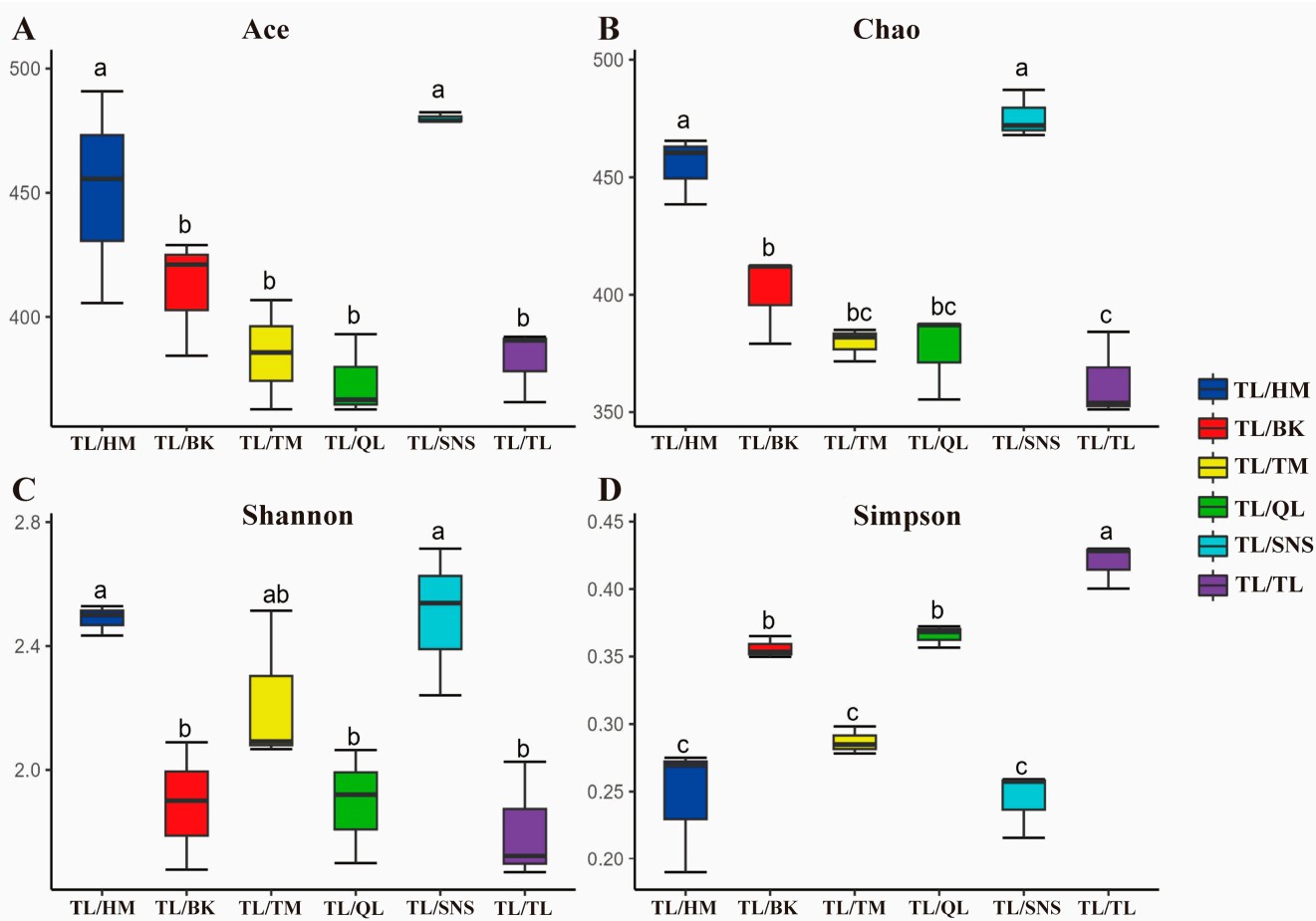

**Figure 6.** The soil fungal diversity index in rhizosphere between grafting and non-grafting plants. (**A**) The Ace indices of soil fungal diversity. (**B**) The Chao indices of soil fungal diversity. (**C**) The Shannon indices of soil fungal diversity. (**D**) The Simpson index of soil fungal diversity. The statistical differences among tissues ($p < 0.05$) are denoted by different lowercase letters. Indicator: TL/HM, Tianlong No. 9/Huimei Zhenba; TL/BK, Tianlong No. 9/Beike; TL/TM, Tianlong No. 9/Torvum; TL/QL, Tianlong No. 9/Qiangli; TL/SNS, Tianlong No. 9/Saint Nise; TL/TL, Tianlong No. 9/Tianlong No. 9.

*3.6. Evaluation of the Impact of Grafting with Various Rootstocks on the Composition of Microbial Communities in Rhizosphere Soil*

In terms of bacterial communities, a total of 41 phyla, 130 classes, 318 orders, 519 families, 1009 genera, 2055 species, and 5672 operational taxonomic units (OTUs) were identified. Similarly, for fungal communities, the analysis revealed eight phyla, 33 classes, 77 orders, 156 families, 198 genera, and 288 species. The study at the phylum level for bacteria revealed significant alterations in the composition of rhizosphere soil bacterial communities following grafting with different rootstocks. The dominant bacterial phyla and their respective proportions across the different treatments included Firmicutes (25.16–43.00%), Proteobacteria (15.91–19.61%), Actinobacteriota (12.81–17.79%), Chloroflexi (8.14–13.43%), Acidobacteriota (3.90–10.01%), Bacteroidota (2.67–3.53%), Myxococcota (2.19–2.79%), Gemmatimonadota (1.84–2.42%), Patescibacteria (0.96–1.23%), and Planctomycetota (0.38–1.10%). Together, these phyla accounted for over 96% of the total relative abundance. Grafting led to changes in the proportions of several phyla, notably increasing the abundance of Actinobacteriota,

Gemmatimonadota, and Planctomycetota, while decreasing the abundance of Firmicutes (Figure 7A).

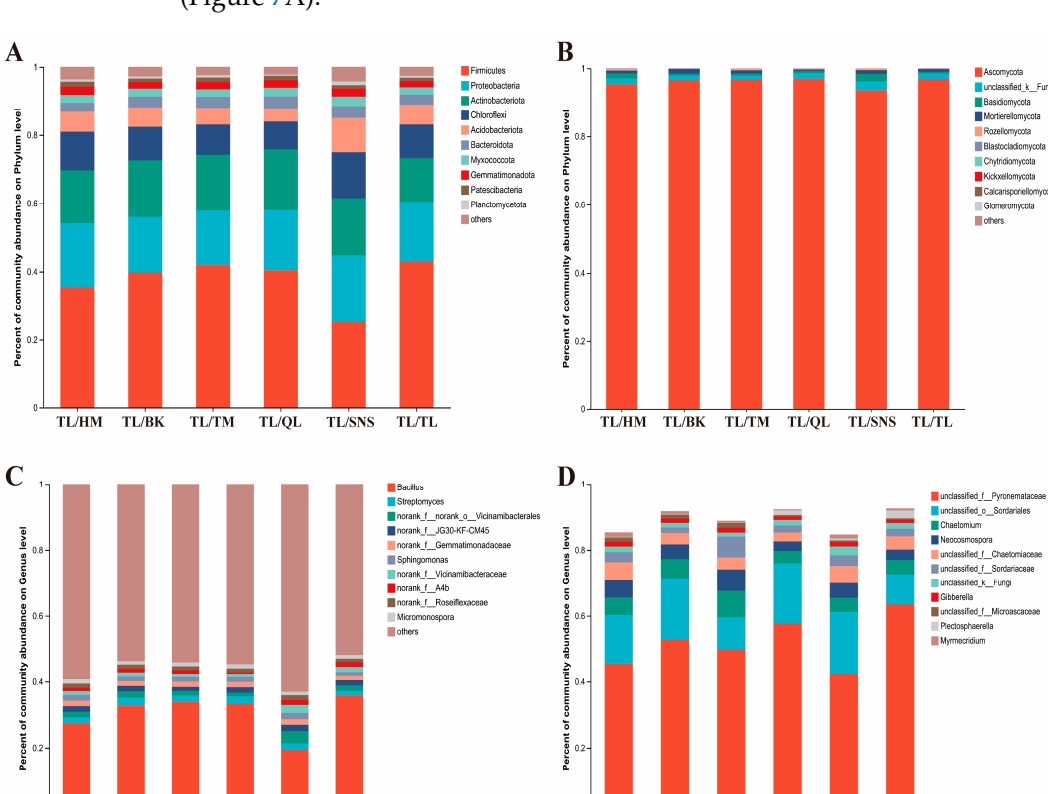

**Figure 7.** The composition of microbial communities in rhizosphere soil after grafting with various rootstocks. (**A**) Relative abundance of bacteria at phylum level. (**B**) Relative abundance of fungi at phylum level. (**C**) Relative abundance of bacteria at genus level. (**D**) Relative abundance of fungi at genus level. Indicator: TL/HM, Tianlong No. 9/Huimei Zhenba; TL/BK, Tianlong No. 9/Beike; TL/TM, Tianlong No. 9/Torvum; TL/QL, Tianlong No. 9/Qiangli; TL/SNS, Tianlong No. 9/Saint Nise; TL/TL, Tianlong No. 9/Tianlong No. 9.

At the phylum level for fungi, the dominant fungal phyla across the treatments were Ascomycota (93.58–96.79%), unclassified_k__Fungi (1.20–2.68%), Basidiomycota (0.50–2.17%), Mortierellomycota (0.43–1.37%), and Rozellomycota (0.10–0.49%). These phyla collectively represented over 99% of the total relative abundance. Grafting particularly increased the abundance of Basidiomycota and Mortierellomycota, especially in the TL/HM and TL/SNS treatments, which showed significant differences compared to the TL/TL self-rooted plants, while the abundance of Ascomycota decreased (Figure 7B).

At the genus level, the analysis of identifiable bacterial communities across different treatments revealed that the dominant bacterial genera were Bacillus, Streptomyces, norank_f__norank_o__Vicinamibacterales, norank_f__JG30-KF-CM45, norank_f__Gemmatimonadaceae, Sphingomonas, norank_f__Vicinamibacteraceae, norank_f__A4b, norank_f__Roseiflexaceae, and Micromonospora, with relative abundances ranging from 19.43% to 35.83%, 1.39% to 2.40%, 1.06% to 3.70%, 1.22% to 1.90%, 1.31% to 1.68%, 1.30% to 1.87%, 0.89% to 2.41%, 0.73% to 1.51%, 1.05% to 1.44%, and 1.00% to 1.36%, respectively. Grafting was found to increase the relative abundance of Streptomyces and Gemmatimonadaceae while decreasing the relative abundance of Bacillus (Figure 7C).

In terms of fungal community composition at the genus level, the analysis across different treatments indicated that the top 11 dominant genera were unclassified_f__Pyronemataceae, unclassified_o__Sordariales, Chaetomium, Neocosmospora, unclassified_f__Chaetomiaceae, unclassified_f__Sordariaceae, unclassified_k__Fungi, Gibberella,

unclassified_f__Microascaceae, Plectosphaerella*, and Myrmecridium, with relative abundances ranging from 42.39% to 63.49%, 9.21% to 18.80%, 3.79% to 7.98%, 2.83% to 6.56%, 2.67% to 5.46%, 1.63% to 6.31%, 1.20% to 2.68%, 1.00% to 1.49%, 0.36% to 1.42%, 0.11% to 1.42%, and 0.29% to 1.27%, respectively. Grafting was observed to increase the relative abundance of unclassified_o__Sordariales, and the eggplant rootstock treatments (TL/HM, TL/BK, and TL/TM) specifically increased the relative abundance of Chaetomium compared to the tomato rootstock treatments (TL/QL and TL/SNS) and the self-rooted plants (TL/TL). Conversely, grafting decreased the relative abundance of unclassified_f__Pyronemataceae and Plectosphaerella (Figure 7D).

*3.7. Venn Diagram Analysis Based on OTU Level*

Venn diagrams are primarily utilized to quantify the number of shared and unique operational taxonomic units (OTUs) across multiple samples. The total number of bacterial OTUs in the rhizosphere soil of different rootstock-grafted plants—TL/HM, TL/BK, TL/TM, TL/QL, and TL/SNS—and self-rooted eggplants (TL/TL) was 2544, 2516, 2516, 2510, 2559, and 2504, respectively. Among these, 2474 OTUs were shared across all treatments, while the number of unique bacterial OTUs was 70, 42, 42, 36, 85, and 30, respectively. The grafted treatments generally exhibited a higher number of unique OTUs compared to the self-rooted plants, with the TL/HM and TL/SNS treatments showing 2.33 and 2.83 times more unique OTUs than the TL/TL treatment, respectively (Figure 8A). For fungal communities, the total number of OTUs in the rhizosphere soil of the different rootstock-grafted plants—TL/HM, TL/BK, TL/TM, TL/QL, and TL/SNS—and self-rooted eggplants (TL/TL) was 358, 294, 320, 307, 350, and 292, respectively (Figure 8B). Of these, 239 OTUs were shared among all treatments, while the number of unique fungal OTUs was 119, 55, 81, 68, 111, and 53, respectively. Similar to bacterial OTUs, the grafted treatments displayed a higher number of unique OTUs compared to the self-rooted plants, with the TL/HM and TL/SNS treatments showing 2.25 and 2.09 times more unique OTUs than the TL/TL treatment, respectively. These findings suggest that grafting increases the number of unique dominant bacterial and fungal species in the rhizosphere soil of plants. The enhanced richness and diversity of these unique dominant microbial species may be a key mechanism through which grafted plants improve their resistance.

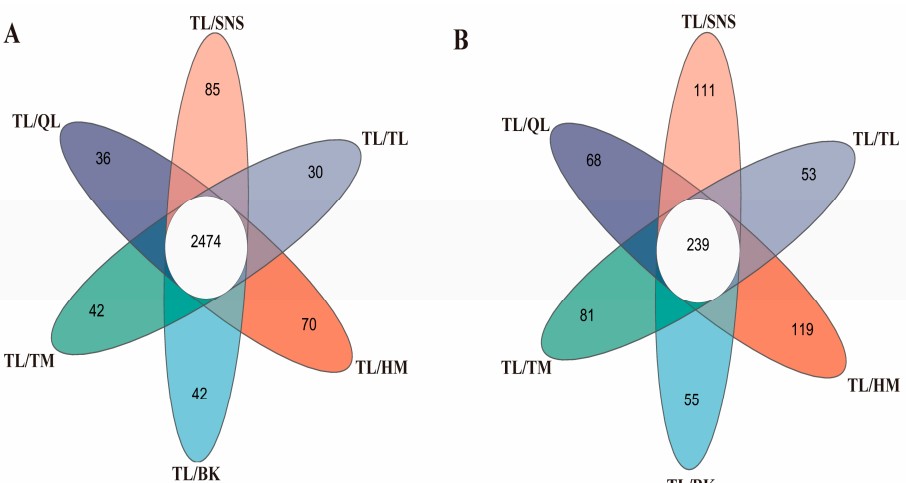

**Figure 8.** Venn diagram analysis based on OTU level after grafting with various rootstocks. (**A**) The total number of bacterial OTUs in the rhizosphere soil of different rootstock-grafted plants. (**B**) The total number of fungal OTUs in the rhizosphere soil of different rootstock-grafted plants. Indicator: TL/HM, Tianlong No. 9/Huimei Zhenba; TL/BK, Tianlong No. 9/Beike; TL/TM, Tianlong No. 9/Torvum; TL/QL, Tianlong No. 9/Qiangli; TL/SNS, Tianlong No. 9/Saint Nise; TL/TL, Tianlong No. 9/Tianlong No. 9.

### 3.8. Correlation Analysis Between Growth Traits, Disease Resistance, Soil Microbial Biomass, Microbial Diversity, and Soil Enzyme Activity in Grafted Eggplants

Redundancy analysis (RDA) was performed using the growth traits of grafted eggplants—plant height, stem diameter, yield, and disease incidence—as response variables, and soil microbial biomass, microbial diversity, and soil enzyme activity as explanatory variables (Figure 9). The results demonstrated a significant relationship between the response and explanatory variables ($p = 0.034$), with the first two RDA components (RDA1 and RDA2) accounting for 61.24% and 19.33% of the total variance in soil microenvironment factors, respectively. Disease incidence was significantly positively correlated with the bacterial Simpson index (Simpson_b) ($r = 0.5466$) and significantly negatively correlated with microbial biomass nitrogen (N), microbial biomass carbon (C), β-glucosidase (BG), acid phosphatase (AP), leucine aminopeptidase (LA), bacterial Ace index (Ace_b), fungal Ace index (Ace_f), and fungal Shannon index (Shannon_f). Conversely, yield, plant height, and stem diameter exhibited significant negative correlations with Simpson_b and significant positive correlations with soil microbial biomass (C and N), soil enzyme activity (BG, AP, and LA), Ace index, and Shannon index.

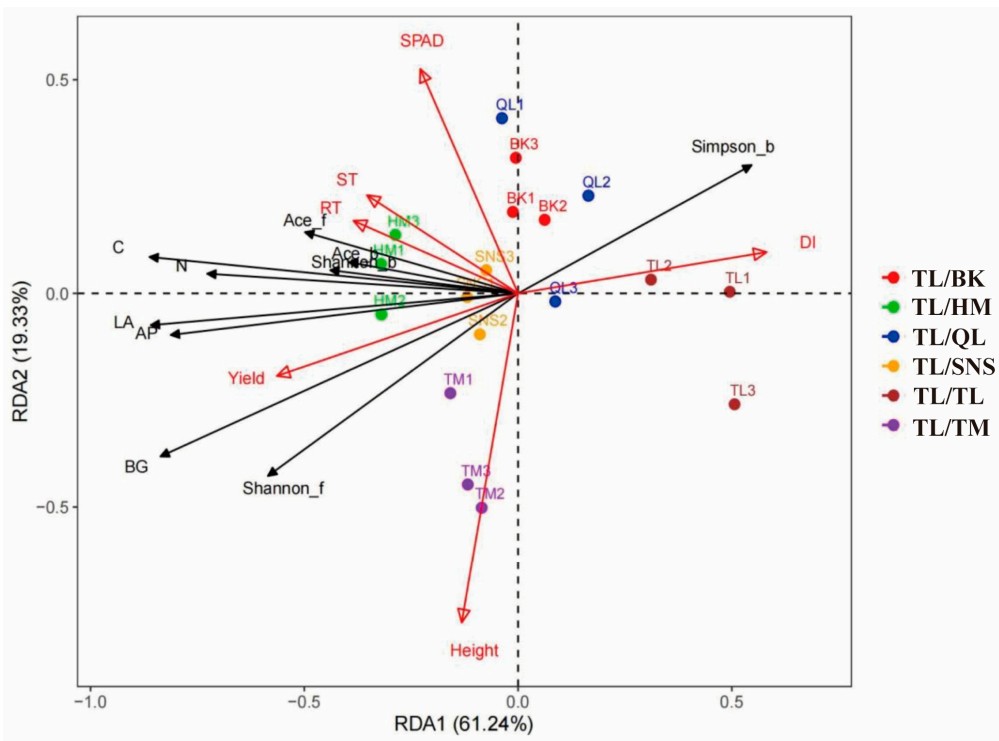

**Figure 9.** The correlation analysis of ten key environmental factors—microbial nitrogen (N), microbial carbon (C), β-glucosidase (BG), acid phosphatase (AP), leucine aminopeptidase (LA), bacterial Ace index (Ace_b), bacterial Shannon index (Shannon_b), bacterial Simpson index (Simpson_b), fungal Ace index (Ace_f), and fungal Shannon index (Shannon_f)—reveals a significant influence ($p = 0.034$) on various growth and health parameters of eggplants, including plant height (Height), rootstock stem diameter (RT), scion stem diameter (ST), yield (Yield), and disease incidence (DI). Indicator: TL/HM, Tianlong No. 9/Huimei Zhenba; TL/BK, Tianlong No. 9/Beike; TL/TM, Tianlong No. 9/Torvum; TL/QL, Tianlong No. 9/Qiangli; TL/SNS, Tianlong No. 9/Saint Nise; TL/TL, Tianlong No. 9/Tianlong No. 9.

As illustrated in the Pearson correlation heat map (Figure 10), disease incidence in grafted eggplant (DI) shows a highly significant positive correlation with the bacterial Simpson index (Simpson_b). In contrast, DI has extremely significant negative correlations with microbial biomass nitrogen (N), microbial carbon (C), β-glucosidase (BG), acid phosphatase (AP), leucine aminopeptidase (LA), bacterial Shannon index (Shannon_b),

fungal Ace index (Ace_f), and fungal Shannon index (Shannon_f). Plant yield is significantly positively correlated with microbial biomass nitrogen (N), microbial carbon (C), β-glucosidase (BG), acid phosphatase (AP), leucine aminopeptidase (LA), and the fungal Shannon index (Shannon_f). It also shows a highly significant positive correlation with the bacterial Shannon index (Shannon_b) and the fungal Ace index (Ace_f). Conversely, yield is highly negatively correlated with the bacterial Simpson index (Simpson_b). Scion stem thickness (ST) shows a highly significant positive correlation with microbial carbon (C) and acid phosphatase (AP), and a significant positive correlation with microbial biomass nitrogen (N), leucine aminopeptidase (LA), and β-glucosidase (BG). ST also has a significant negative correlation with the bacterial Simpson index (Simpson_b). Rootstock stem diameter (RT) is significantly positively correlated with β-glucosidase (BG). Additionally, plant height shows significant positive correlations with both β-glucosidase (BG) and the fungal Shannon index (Shannon_f).

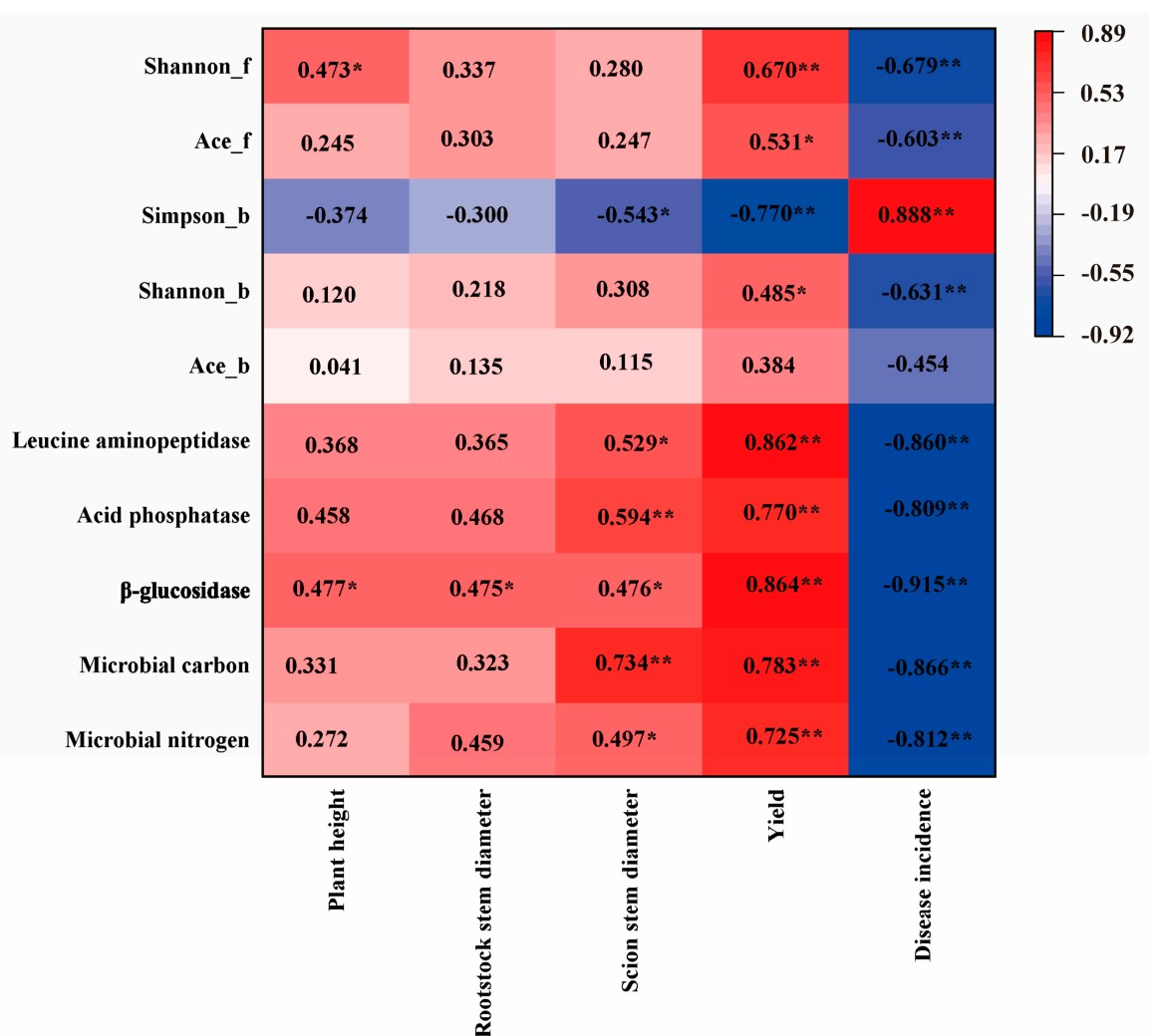

**Figure 10.** Correlation analysis of growth traits of grafted eggplants with soil microecological factors and other indicators. Differences were considered significant at * $p < 0.05$ and highly significant at ** $p < 0.01$.

## 4. Discussion

Extensive research and practical applications have demonstrated that grafting onto different rootstocks can result in varying degrees of growth vigor and yield, and control bacterial wilt disease in grafted plants [35,36]. For instance, grafted eggplants generally exhibit increased root length (63.65 cm) and yield (6.69 kg/plant) compared to self-rooted

plants. Similarly, grafted tomatoes show significantly higher yields and improved nutrient uptake, including nitrogen (25.67%), phosphorus (20.41%), and potassium (20.97%) compared to self-rooted tomatoes [37]. Grafted cucumber seedlings also benefit from enhanced root development, a higher root-to-shoot ratio, and increased chlorophyll content in leaves compared to non-grafted seedlings [38]. In this study, eggplants grafted onto five different rootstocks exhibited an average yield increase of 25.19%. Specifically, the TL/HM, TL/TM, and TL/SNS combinations showed yield increases of 36.89%, 30.38%, and 26.43%, respectively, compared to self-rooted eggplants. Additionally, grafted plants had significantly larger stem diameters and higher leaf chlorophyll content, with the TL/HM treatment reaching the highest chlorophyll content of 54.23 ± 3.17 SPAD, a value significantly greater than that of self-rooted plants. Furthermore, grafting significantly reduced the incidence of bacterial wilt. The lowest disease incidences were recorded in the TL/HM (3.33%), TL/SNS (5.56%), and TL/TM (7.78%) treatments, whereas self-rooted eggplants experienced a much higher incidence of 55.56%. These results suggest that in fields with continuous cropping or disease-prone soils, the use of high-quality rootstocks for grafting can significantly enhance disease resistance and promote consistent yield increases. Overall, the analysis indicates that grafting eggplants onto eggplant-type rootstocks is generally more effective than using tomato-type rootstocks.

Plant resistance to soil-borne diseases is intricately connected to the composition of rhizosphere microorganisms [39]. Bacteria, fungi, and actinomycetes are the three predominant groups of soil microorganisms, and their abundance and diversity reflect biological activity in the soil, thereby indicating the intensity of nutrient cycling and material metabolism [40]. Soil microbial communities and diversity are key regulators of soil ecosystem multifunctionality, influencing soil resilience and disease resistance [41–43]. Among these groups, bacteria constitute 70% to 90% of the total microbial population, playing key roles in maintaining soil health, suppressing soil-borne pathogens, and regulating nutrient cycling by controlling nutrient flow and exchange processes [44,45]. Fungi contribute to the breakdown of complex organic compounds such as carbohydrates, cellulose, lignin, and tannins, and are involved in humus formation, ammonification, and nitrification [46]. Actinomycetes are important for nitrogen fixation, phosphorus solubilization, organic acid production, and the synthesis of indole-3-acetic acid (IAA), all of which are crucial for regulating microbial abundance, promoting plant growth, and controlling soil-borne diseases [47]. Increasing microbial diversity in the soil has been shown to effectively reduce pathogen infections and enhance plant health [48].

The results of this experiment showed that, compared to self-rooted plants, grafted eggplants had a higher abundance of *Actinobacteriota*, *Streptomyces*, and *Chaetomium* in the rhizosphere soil. Actinomycetes play a crucial role in the decomposition of soil organic matter and the production of antibiotics that combat soil-borne pathogens. *Streptomyces*, the largest genus within the *Actinobacteriota* phylum, is known for its beneficial properties in agriculture, often serving as a biocontrol agent and biostimulant [49]. Around 50% of *Streptomyces* species can produce antibiotics, including widely used ones such as streptomycin, kanamycin, mitomycin, oxytetracycline, kasugamycin, polyoxin, jinggangmycin, and nisin. In addition to antibiotic production, *Streptomyces* produces enzymes like chitinase, glucanase, cellulase, lipase, and nuclease, which inhibit the growth of pathogens [50,51]. Certain species of *Streptomyces* have been shown to suppress the proliferation of *Ralstonia solanacearum* by disrupting its cell membranes, effectively controlling bacterial wilt [52]. Moreover, OTU-level analysis revealed that grafting not only increased the overall number of bacterial and fungal OTUs in the rhizosphere but also boosted the number of unique OTUs. These findings suggest that grafting can significantly modify the rhizosphere microbial community structure by increasing the abundance of beneficial microorganisms such as *Actinobacteriota*, *Streptomyces*, and *Chaetomium*. This enhancement likely promotes the activity of antagonistic microorganisms, thereby helping to restore the balance of the soil microbial community disrupted by continuous cropping and improving the disease resistance of grafted plants compared to self-rooted plants.

Soil microbial biomass represents the total quantity of microorganisms present in the soil, including bacteria, fungi, and actinomycetes. It is composed primarily of microbial biomass carbon, nitrogen, phosphorus, and sulfur. Changes in microbial biomass are highly sensitive indicators of alterations in soil nutrient dynamics, providing valuable insights into the intensity of organic matter decomposition [53,54]. An increase in microbial biomass typically signals an enhanced capacity of the soil to mineralize organic matter, thus facilitating the provision of essential nutrients required for plant growth and yield formation [55]. Microbial biomass carbon is a key component, serving as a critical carbon source that reflects the soil's physicochemical properties and microbial activity, and plays an essential role in maintaining ecological balance [56,57]. Microbial biomass nitrogen, on the other hand, is an important indicator of nitrogen availability in the soil and is essential for the nitrogen cycle and its supply to plants [58]. Both microbial biomass carbon and nitrogen are strongly correlated with soil organic matter content, making them important indicators of overall soil fertility [59,60].

In this study, we observed that microbial biomass levels of carbon, nitrogen, and phosphorus in the rhizosphere soil of grafted eggplants were significantly higher compared to those in self-rooted plants. These results indicate that the interaction between rootstock and scion in grafted plants plays a critical role in increasing microbial biomass within the rhizosphere. This observation aligns with the findings of Pang et al. [17], who reported similar enhancements in soil microbial biomass in tomatoes grafted onto eggplant rootstocks. Grafting modifies the rhizosphere's microbial biomass, promoting nutrient activity and ensuring a more efficient nutrient supply, which in turn contributes to improved crop resistance. Therefore, beyond rootstock-scion compatibility, the inherent resistance characteristics of the rootstock should be a primary consideration. The microbial biomass profile in the rhizosphere of eggplants grafted onto both eggplant- and tomato-type rootstocks may provide a useful indicator for assessing the resistance strength of the rootstock, although further research is needed to confirm these findings.

Soil enzymes, as some of the most biologically active components in soil, play essential roles in nearly all nutrient and organic matter cycling processes [61]. They are key indicators for evaluating soil quality in terrestrial ecosystems and are often used to assess soil fertility [62]. Research has demonstrated that elevated soil enzyme activity is linked to increased biological activity within the soil, which can enhance plant resistance to diseases [63,64]. A strong positive correlation between soil enzyme activity and disease resistance has been widely documented. In this study, the activities of enzymes involved in the carbon, nitrogen, and phosphorus cycles—specifically β-glucosidase, aminopeptidase, and phosphatase—were significantly higher in the rhizosphere of grafted eggplants compared to self-rooted plants. These results suggest that grafting not only enhances the ecological health of the rhizosphere but also strengthens the resistance of grafted plants by boosting enzyme activity in the soil.

## 5. Conclusions

In summary, this study aimed to identify the optimal rootstock and scion combination to enhance the growth, yield, and disease resistance of eggplants by investigating both eggplant-type and tomato-type rootstocks. The research evaluated key growth traits, including stem diameter, plant height, and chlorophyll content, and analyzed microbial biomass and enzyme activity in the rhizosphere soil. The findings demonstrated that grafting eggplants onto eggplant-type rootstocks, particularly 'Huimei Zhenba' (HM) and 'Torvum' (TM), increased growth parameters, including stem diameter and chlorophyll content, and yield. These grafted plants demonstrated superior disease resistance (TL/HM: 8.33%), reducing bacterial wilt incidence (TL/HM: 3.33%) compared to self-rooted controls (TL/TL: 55.56%). Additionally, grafting notably enriched the rhizosphere soil with beneficial microbial communities, enhancing microbial biomass and enzyme activities associated with carbon, nitrogen, and phosphorus cycling. Specifically, grafted eggplants exhibited higher abundances of beneficial microbes, such as *Streptomyces* (35.83%) and *Chaetomium*

(63.49%), known for their pathogen-suppressing properties. This microbiome shift in the rhizosphere likely fosters an environment conducive to improved nutrient availability and sustained soil health. These results provide practical insights for selecting highly resistant rootstock varieties and highlight the importance of rootstock-scion compatibility in sustainable crop production. Future research is recommended to further explore the physiological and molecular mechanisms behind the beneficial interactions between rootstocks and rhizosphere microorganisms.

**Author Contributions:** Writing—original draft preparation, formal analysis, G.D.; writing—review and editing, D.Z.; data curation, H.H.; data curation, X.L.; project administration, Y.Y.; supervision writing—review and editing, Z.Q. All authors have read and agreed to the published version of the manuscript.

**Funding:** This research was funded by Hainan Province Science and Technology Special Fund, grant number ZDKJ2021005; the earmarked fund for HVARS, grant number HNARS-05-Z02; and the Field Soil Scientific Research Station in Danzhou of Hainan Province.

**Data Availability Statement:** The original contributions presented in the study are included in the article, further inquiries can be directed to the corresponding author.

**Conflicts of Interest:** The authors declare no conflicts of interest.

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
