# Peer review of "The Impact of Grafting with Different Rootstocks on Eggplant (Solanum melongena L.) Growth and Its Rhizosphere Soil Microecology"

_agronomy, doi:10.3390/agronomy14112616_

Round 1
Reviewer 1 Report (Previous Reviewer 2)
Comments and Suggestions for Authors
Interesting experience with a lot of chemical analyses. Minor methodological shortcomings shown in the text of the publication. It is not known how many plants there were in one variety and rootstock treatment (30 or 12), were there any repetitions, were soil samples taken in one place or was it a mixed sample taken from different places, how many? In the introduction, provide the values ​​obtained by other authors using rootstocks, how they influenced growth, yield, resistance to soil pathogens, and what the changes were compared to the control.

Author Response
Point-by-point response to reviewers' comments
Reviewer 1:
Interesting experience with a lot of chemical analyses. Minor methodological shortcomings shown in the text of the publication. It is not known how many plants there were in one variety and rootstock treatment (30 or 12), were there any repetitions, were soil samples taken in one place or was it a mixed sample taken from different places, how many? In the introduction, provide the values obtained by other authors using rootstocks, how they influenced growth, yield, resistance to soil pathogens, and what the changes were compared to the control.
Author response: Thank you for your questions and constructive comments. Regarding the replication details, each treatment consisted of 30 plants of a given cultivar arranged in a 9-square-meter plot, which was considered one replicate. Each treatment had three replicates, resulting in a total of 90 plants (3 replicates × 30 plants per replicate). For the evaluation of growth traits, 12 plants were assessed per treatment group. These 12 plants were selected from each plot (which contained 30 plants) using an S-type sampling method. Concerning the soil sampling, field soil samples were collected using the S-type sampling method, with three soil samples taken per treatment. Additionally, in response to your comments on the introduction, we have revised and included values obtained by other authors using rootstocks. These additions have been incorporated into the revised manuscript.
Reviewer 2 Report (New Reviewer)
Comments and Suggestions for Authors
Summary
The authors provided a thorough assessment of microbial community variations linked to different eggplant rootstocks, with results that are well-organized and clearly presented. The manuscript is nearly ready for acceptance, pending a few revisions, particularly in the Materials and Methods section, which would benefit from clearer organization—consider dividing it into subsections like "Experimental Design" , "Environmental Conditions" and “Data analysis” for an improved readability. Additionally, in the introduction section, to enhance the discussion on rootstock-scion interactions, integrating recent references on interspecific rootstock-scion combinations would contextualize the significance of rootstock selection in optimizing microbial communities. Including images of the field setup, plants, and rootstocks would further provide readers with helpful visual context. Finally, if feasible, adding morphometric data for rootstocks, such as root diameter, root length, or root mass, would offer more insight into rootstock effects on the microbial environment.
Title
Ensure the title adheres to MDPI’s guidelines by capitalizing the first letter of each significant word. Additionally, please include the Latin name for eggplant (Solanum melongena L.) in brackets.
Abstract
- Begin the abstract with a concise background statement on the importance of microbial communities in relation to rootstock performance (1-2 lines).
- Enhance the readability of lines 12-13 by clarifying transitions and improving sentence flow.
- Briefly mention the trial’s specific environment (e.g., greenhouse or open field) and the growing period, as these conditions are integral to understanding microbiome variations.
Introduction
The introduction would benefit from additional references highlighting how rootstock performance varies under different environmental conditions, especially considering that interspecific rootstock use is an effective approach for improving eggplant crop resilience. In this context, I highly recommend integrating the following recent and relevant references to support the discussion of rootstock adaptability and performance: https://doi.org/10.3390/horticulturae9091060; https://doi.org/10.1080/14620316.2023.2300721; https://doi.org/10.1080/14620316.2023.2300721; https://doi.org/10.3390/agronomy13112705)
Materials and Methods
The plant material section is thorough and informative.
- Suggest renaming subsection 2.2 as “Experimental Design” and moving all environmental condition details to a new subsection titled “Environmental Conditions.”
- Specify where environmental parameters were sourced (e.g., online site or specific sensor) and, if possible, provide a graph showing temperature and precipitation trends (e.g., by decade).
- Statistical analysis should be also moved in a separate subsection.
Results
- Rootstock Morphometrics: Did you perform any biomorphometric analysis of the rootstocks (e.g., main root diameter, lateral root diameter, root system width, root weight)? If so, consider including these measurements.
- Qualitative Analysis of Fruits: If you conducted a qualitative fruit analysis or have high-quality images of the fruits, consider including them for added detail.
- Soil Analysis: Did you conduct a complete soil analysis at the start and end of the trial? Details such as cation exchange capacity, pH, and other soil parameters could clarify how the rootstock impacted microbial activity. If available, please include this information.
- Pearson’s Correlation: If you have not performed Pearson’s correlation analysis, consider including it to better elucidate trait correlations. A graphical representation of the correlations (e.g., using heatmaps or correlation plots) could enhance readers' understanding.
Discussion
The discussion effectively explores and contextualizes the findings. A few additional points to consider:
- Did you encounter nematodes during your study? Are they known to be present in your growing area, and if so, were the rootstocks employed resistant to them?
Author Response
Point-by-point response to reviewers' comments
Reviewer 2:
The authors provided a thorough assessment of microbial community variations linked to different eggplant rootstocks, with results that are well-organized and clearly presented. The manuscript is nearly ready for acceptance, pending a few revisions, particularly in the Materials and Methods section, which would benefit from clearer organization—consider dividing it into subsections like "Experimental Design" , "Environmental Conditions" and “Data analysis” for an improved readability. Additionally, in the introduction section, to enhance the discussion on rootstock-scion interactions, integrating recent references on interspecific rootstock-scion combinations would contextualize the significance of rootstock selection in optimizing microbial communities. Including images of the field setup, plants, and rootstocks would further provide readers with helpful visual context. Finally, if feasible, adding morphometric data for rootstocks, such as root diameter, root length, or root mass, would offer more insight into rootstock effects on the microbial environment.
Title
Ensure the title adheres to MDPI’s guidelines by capitalizing the first letter of each significant word. Additionally, please include the Latin name for eggplant (Solanum melongena L.) in brackets.
Author response: Thank you for your constructive feedback. We have revised the manuscript title to align with MDPI’s guidelines and have included the Latin name for eggplant (Solanum melongena L.) accordingly.
Abstract
- Begin the abstract with a concise background statement on the importance of microbial communities in relation to rootstock performance (1-2 lines).
- Enhance the readability of lines 12-13 by clarifying transitions and improving sentence flow.
- Briefly mention the trial’s specific environment (e.g., greenhouse or open field) and the growing period, as these conditions are integral to understanding microbiome variations.
Author response: Thank you for your constructive comment. We have revised the abstract in accordance with your suggestions.
Introduction
The introduction would benefit from additional references highlighting how rootstock performance varies under different environmental conditions, especially considering that interspecific rootstock use is an effective approach for improving eggplant crop resilience. In this context, I highly recommend integrating the following recent and relevant references to support the discussion of rootstock adaptability and performance: https://doi.org/10.3390/horticulturae9091060; https://doi.org/10.1080/14620316.2023.2300721; https://doi.org/10.3390/agronomy13112705)
Author response: Thank you for your constructive comment. We have integrated the references you suggested to enhance and support the discussion on rootstock adaptability and performance in the manuscript. However, we were unable to access the article titled ‘Grafting as an Effective Approach for Improvement of Eggplant Resistance Against Bacterial Wilt (Ralstonia solanacearum) in Screen-House Condition’ (https://doi.org/10.1080/14620316.2023.2300721).
Materials and Methods
The plant material section is thorough and informative.
Suggest renaming subsection 2.2 as “Experimental Design” and moving all environmental condition details to a new subsection titled “Environmental Conditions.”
Author response: Thank you for your constructive suggestion. We have renamed subsection 2.2 to ‘Experimental Design’ and moved all environmental condition details to a new subsection, titled 2.3 ‘Environmental Conditions’.
Specify where environmental parameters were sourced (e.g., online site or specific sensor) and, if possible, provide a graph showing temperature and precipitation trends (e.g., by decade).
Author response: Thank you for your constructive comment. The experimental site, Danzhou Caitian Field Observation and Experimental Station of the Chinese Academy of Tropical Agricultural Sciences was established in 2016 and is managed by the Hainan Provincial Department of Science and Technology. It is equipped with a small climate station for long-term monitoring. Below, we provide a table summarizing the average annual temperature (°C) and annual rainfall (mm) at the experimental site from 2016 to 2022, which has also been included as both a table and graph in the manuscript.
Table 1. The average annual temperature (°C) and annual rainfall (mm) at the experimental site were recorded from 2016 to 2022 at the Danzhou Caitian Field Observation and Experimental Station.
Year |
2016 |
2017 |
2018 |
2019 |
2020 |
2021 |
2022 |
Average annual temperature (°C) |
24.7 |
24.55 |
24.52 |
25.38 |
25.58 |
24.96 |
24.00 |
Annual rainfall (mm) |
1831.0 |
2164.0 |
1525.0 |
1293.5 |
1180.1 |
1666.9 |
1783.2 |
Statistical analysis should be also moved in a separate subsection.
Author response: Thank you for your constructive suggestion. We have moved the statistical analysis to a new subsection titled ‘2.4 Statistical Analysis’.
Results
Rootstock Morphometrics: Did you perform any biomorphometric analysis of the rootstocks (e.g., main root diameter, lateral root diameter, root system width, root weight)? If so, consider including these measurements.
Author response: Thank you for your constructive comment. In this study, we only measured the stem diameter of the rootstock and did not analyze other root-related indicators. Your suggestion will enable us to gain a more comprehensive understanding of root morphology across different rootstocks, which will be highly beneficial for our future research. Thank you again for your valuable suggestion.
Qualitative Analysis of Fruits: If you conducted a qualitative fruit analysis or have high-quality images of the fruits, consider including them for added detail.
Author response: Thank you for your constructive comment. This experiment did not include a qualitative analysis of the fruits. However, we plan to address the aspects you mentioned, including fruit morphology and quality, in our future work. We appreciate your valuable suggestions.
Soil Analysis: Did you conduct a complete soil analysis at the start and end of the trial? Details such as cation exchange capacity, pH, and other soil parameters could clarify how the rootstock impacted microbial activity. If available, please include this information.
Author response: Thank you for your constructive comment. Soil properties were analyzed prior to the experiment; however, we did not conduct soil analysis post-experiment. Your suggestion is highly appreciated and will be instrumental in enhancing future experimental designs.
Pearson’s Correlation: If you have not performed Pearson’s correlation analysis, consider including it to better elucidate trait correlations. A graphical representation of the correlations (e.g., using heatmaps or correlation plots) could enhance readers' understanding.
Author response: Thank you for your constructive comment and suggestion. In response, we have generated and included a Pearson correlation heat map to present the relationships between factors more clearly and intuitively.
Discussion
The discussion effectively explores and contextualizes the findings. A few additional points to consider:
Did you encounter nematodes during your study? Are they known to be present in your growing area, and if so, were the rootstocks employed resistant to them?
Author response: Thank you for your constructive comment. At our experimental site and growing area, nematodes were not observed, as they are not present in this growing area. Consequently, there was no nematode infection during our trials. However, your suggestion is highly valuable, and in future work, we plan to assess the rootstock's resistance to nematodes. This will help us identify rootstocks that are resistant to both bacterial wilt and nematodes. We appreciate your insightful suggestion.
Reviewer 3 Report (New Reviewer)
Comments and Suggestions for Authors
Dear editor;
I evaluated the article. The article is generally well designed, written and written. It is an original article that includes modern analyses. The references used are relevant to the subject. It will make significant contributions to the literature. However, there are some points that need to be corrected.
Find;
Add the Latin name of eggplant to the title.
Write the full name instead of using abbreviations in the abstract section.
Also, there is a lot of numerical data in the study. Why didn't you add the prominent data to the abstract?
There may be many parameters affecting yield and quality. Write these in a few detailed paragraphs in the introduction (genetics, ecology, maintenance conditions, etc.)
A photo from the experimental site can be added..
Add more references in the method section.
The data in the study is presented with graphics. I think it is sufficient. However, some can be tabulated.
The discussion and conclusion sections are sufficient. It is well written.
Finally, I think the conclusion section should be written in more detail.
Author Response
Point-by-point response to reviewers' comments
Reviewer 3:
Add the Latin name of eggplant to the title.
Author response: Thank you for your constructive comment. The Latin name of eggplant has been added to the title.
Write the full name instead of using abbreviations in the abstract section.
Author response: Thank you for your constructive comment. We have revised the abstract and removed the abbreviations.
Also, there is a lot of numerical data in the study. Why didn't you add the prominent data to the abstract?
Author response: Thank you for your constructive comment. Based on your suggestion, we have added key data from the results section to the abstract.
There may be many parameters affecting yield and quality. Write these in a few detailed paragraphs in the introduction (genetics, ecology, maintenance conditions, etc.)
Author response: Thank you for your constructive comment. We have added a paragraph discussing additional factors affecting yield and quality, including genetics and cultivation conditions.
A photo from the experimental site can be added.
Author response: Thank you for your constructive suggestion. We have added photos of the different graft combinations at the experimental site—TL/TM (Tianlong No. 9/Torvum), TL/HM (Tianlong No. 9/Huimei Zhenba), TL/BK (Tianlong No. 9/Beike), TL/TL (Tianlong No. 9/Tianlong No. 9), TL/QL (Tianlong No. 9/Qiangli), and TL/SNS (Tianlong No. 9/Saint Nise)—to the manuscript.
Add more references in the method section.
Author response: Thank you for your constructive comment. We have added additional references to the Methods section.
The data in the study is presented with graphics. I think it is sufficient. However, some can be tabulated.
Author response: Thank you for your constructive comment. We have converted Figures 3 and 4 into tables: Table 2 now presents the soil microbial biomass analysis for carbon, nitrogen, and phosphorus, while Table 3 details the soil enzyme activity analysis for β-glucosidase, aminopeptidase, and phosphatase.
The discussion and conclusion sections are sufficient. It is well written.
Author response: Thank you for your positive feedback.
Finally, I think the conclusion section should be written in more detail.
Author response: Thank you for your constructive comment. We have revised the conclusion based on your suggestion.
Round 2
Reviewer 2 Report (New Reviewer)
Comments and Suggestions for Authors
Authors thoroughly addressed to all the comments. The new Heatmap provided is amazing.
This manuscript is a resubmission of an earlier submission. The following is a list of the peer review reports and author responses from that submission.
Round 1
Reviewer 1 Report
Comments and Suggestions for Authors
The manuscript entitled ‘The impact of grafting with different rootstocks on eggplant growth and its rhizosphere soil microecology’ fits within the general scope of Agronomy MDPI. The authors appraised about eggplant grafting on solanaceous rootstocks, in comparison to self-rooted eggplant and also highlighted rootstocks associated soil microbial biomass, enzyme activity, and a more diverse microbial community, contributing to the increased resistance and yield.
The manuscript could be significant to the field; however, it contains several flaws.
Major flaws:
- In this experiment, error degrees for freedom seem to be less to compute the experimental error variance and its robustness.
- The experimental data is of single season and not adequately presented for growth and yield attributes. Moreover, a confirmation or validation study is required for concluding the best-performing rootstock in field experiments.
- The manuscript has a high similarity rate (28%, excluding the ‘References’ section).
Minor flaws:
Line #12-14: The sentence is too long; it shall be fragments into two separate sentences.
Line #62-64: This sentence should be reframed. It is tomato grafted onto eggplant rootstock; not eggplant rootstocks grafted onto tomato.
Line #96: Italicized the scientific name of Solanum torvum
Line #101: The experiment was conducted for a single crop growing season (October 2021 to May 2022). How it can be reliable to support the conclusion since the performance of grafted plants may vary across seasons and years in the field evaluation where climactic variables are non-controllable.
Line #105-107: The standard way of depicting the graft combinations is first to mention scion and then rootstocks (for instance, Tianlong No. 9/Huimei Zhenba or for figures TL/HM). It should be presented in this way throughout MS.
Line #108-110: Which field experimental design was employed while transplanting seedlings? And considering the treatments and replication, the error degrees for freedom seem to be less to compute the experimental error variance.
Line #117-118: What was the level of bacterial (Ralstonia solanaceraum) load/CFU already present in the soil before transplanting of grafts? Whether it was quantified? Since the aim of the study was to address the issues of bacterial wilt using grafting and its associated microbiota.
Line #233-239: Considering the treatments and replications in the present experiment, the error degrees for freedom seem to be less to compute the experimental error variance.
Line #276: How was scoring done for bacterial wilt-infected plants? Mention the scoring method in the material and methods section.
Line #280: The figure 2 depicts the bacterial wilt incidence is 70% in non-grafted (TL), and here is it 55.56%?
Line #282: Yield is highly dependent on traits, and data pertaining to its attribute traits like fruit number and fruit weight should be presented, which is missing in the MS. The yield should be presented in standard format and per unit area; either it should be yield per plant or kg per square meter.
Line #372: What does this mean here, “This section is not mandatory but may be added if there are patents resulting from the work reported”?
Line #432: Figures 7A-D are not clearly visible, presented for composition of microbial 388 communities in rhizosphere soil.
Line #540: All scientific names should be in italics through the MS.
For the aforesaid considerations, I consider the manuscript not suitable for publication in the Agronomy MDPI journal.
Reviewer 2 Report
Comments and Suggestions for Authors
An interesting experience with a large number of biochemical and physiological analyzes performed. In the introduction, remove the final paragraph and state the purpose of the research and its justification. Appropriate number of treatments and repetitions, their number should be provided for laboratory analyses. The methodology does not describe how growth and yield measurements were made. Some fragments of the results description (marked in yellow) should be removed. Incorrect choice of words regarding the height and diameter of the plants. Charts, especially the description of the X axis and the units are too small and not legible, improve the font size. In the discussion, cite results obtained by other authors, which facilitates interpretation for the reader. The number of publications used and their compliance with the research topic are correct.

Some parts of the description of the results should be corrected in terms of the use of inappropriate or imprecise terms in English, e.g. height and diameter.